# Thinking vs. Doing: Improving Agent Reasoning by Scaling Test-Time Interaction

**Junhong Shen**[1,2*]  **Hao Bai**[3*]  **Lunjun Zhang**[4]  **Yifei Zhou**[5]  **Amrith Setlur**[1]

**Shengbang Tong**[7]  **Diego Caples**[6]  **Nan Jiang**[3]  **Tong Zhang**[3]

**Ameet Talwalkar**[1]  **Aviral Kumar**[1]

[1]CMU  [2]Scribe  [3]UIUC  [4]U of Tronoto  [5]UC Berkeley  [6]The AGI Company  [7]NYU

## Abstract

The current paradigm of test-time scaling relies on generating long reasoning traces ("thinking" more) before producing a response. In agent problems that require interaction, this can be done by generating thinking traces before acting in the world. However, this process does not allow agents to acquire new information from the environment or adapt their behavior over time. In this work, we propose to scale **test-time interaction**, an untapped dimension of test-time scaling that increases the agent's interaction horizon to enable running rich behaviors such as exploration, backtracking, and dynamic re-planning within a single rollout. To demonstrate the promise of this scaling dimension, we study the domain of web agents. We first show that even prompting-based interaction scaling without any training can improve task success on web benchmarks non-trivially. Building on this, we introduce *TTI* (Test-Time Interaction), a curriculum-based online reinforcement learning (RL) approach that trains agents by adaptively adjusting their rollout lengths. Using a Gemma 3 12B model, *TTI* produces state-of-the-art open-source, open-data web agents on WebVoyager and WebArena benchmarks. We further show that *TTI* enables agents to balance exploration and exploitation adaptively. Our results establish interaction scaling as a powerful, complementary axis to scaling per-step compute, offering new avenues for training adaptive agents.

**Project page:** https://test-time-interaction.github.io
**Code:** https://github.com/test-time-interaction/TTI

## 1 Introduction

Recent advances in foundation models have enabled a shift from static language models to interactive agents that perform multi-step tasks in dynamic environments like browsers [1–6], terminals [7], and the physical world [8–13]. These agents operate in closed-loop settings where each action changes the current state of the world and affects future interaction with the environment. As a result, interactive agents must plan under uncertainty and adapt to failures in real time to be successful. How can we build agents that succeed in such interactive settings?

Current post-training approaches produce *reactive* agents that respond to immediate observations but struggle with evolving or uncertain task dynamics. Methods like supervised fine-tuning (SFT) on expert demonstrations [14–18] or reinforcement learning (RL) with task rewards [19–23] typically train agents to predict a *single best action* at each step. Even with test-time scaling, where agents

---

*Equal contribution. Correspondence to: *junhongs@andrew.cmu.edu, haob2@illinois.edu*

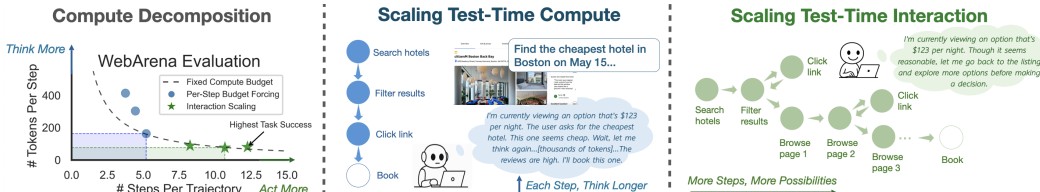

Figure 1: We propose a new-axis of test-time scaling for agents: **scaling the number of interaction steps**. Unlike traditional methods that emphasize longer reasoning per step, we show that acting more helps gain new information from the environment and improve task performance (detailed results of the left plot in Section 4.2).

are prompted to "think" longer before prescribing an action [24–26], they are still optimized to select the most effective action based on the agent's internal state. While sufficient for fully observable and stationary tasks, reactive policies based on the agent's internal estimate of the task state are often sub-optimal in partially observable (e.g., incomplete details visible on a page) or non-stationary (e.g., fluctuating prices during flight booking) settings, where adaptive, information-seeking behavior is critical.

In this paper, we argue that instead of reactive "optimal" policies, agents should learn adaptive policies that can collect new information from the environment and adjust their behaviors on-the-fly. A pre-requisite for such adaptability is the ability to *take more actions* during deployment than those prescribed by an expert trajectory. We therefore propose a new dimension of test-time scaling: *increasing the number of interaction steps of the agent*. This allows agents to have sufficient context and time to attempt different behaviors. For example, in a hotel booking task, an agent must first browse many listings to compare user reviews and check availability before selecting the best option. Interaction scaling is orthogonal to existing methods based on chain-of-thought (CoT), which emphasize deeper reasoning per step but do not support information-gathering from the environment. This notion of information gain is unique to agentic tasks with partial observability and requires interaction, not merely larger per-step compute. For instance, an agent that reasons deeply about one selected hotel without interacting further may miss better options that show up only after exploration.

Although the idea of interaction scaling is conceptually straightforward, extending it to post-training and teaching agents to scale interaction autonomously presents key challenges. Without appropriate training signals, agents may overfit to exploratory behaviors like blindly clicking links but not making progress toward the actual task objective, wasting the additional steps. To tackle this issue, we propose to combine online RL with a curriculum that prescribes how to scale the interaction horizon, training agents that first learn effective exploitation before extending their horizon to explore.

We instantiate our approach in the domain of web agents, a widely applicable setting with well-established benchmarks. We first show that scaling test-time interaction via prompting the agent to "think and act again" after it decides to terminate can already improve the task success rate from 23% to ≥ 28% on WebArena [2] (see Figure 3 for details). While this increases trajectory length and the number of tokens generated, spending an equivalent amount of compute on conventional test-time scaling methods like forcing the agent to think for longer [27] or running best-of-$n$ [28–30] yields less than a 3% gain. These findings validate the potential of interaction scaling.

We then move beyond prompting and develop **TTI** (Test-Time Interaction), a curriculum-based RL approach that trains agents to adaptively scale interaction by gradually increasing the rollout horizon. We scale **TTI** to large and diverse training sets (>100K tasks across ∼20 domains) by integrating an automated task generation pipeline [19]. **TTI** achieves state-of-the-art performance among open-source agents trained on open data on both WebVoyager [1] and WebArena [2], using only a 12B Gemma 3 model, improving over the non-fine-tuned agent by 9% and 8%, respectively. Our analysis further shows that curriculum training enables adaptive exploration: agents learn to initiate new searches or backtrack in complex tasks, while following efficient paths in simpler ones.

**In summary**, we introduce interaction scaling as a new dimension for test-time scaling of agents. We propose **TTI** to train agents by adjusting interaction horizon dynamically. Our results show that **TTI** yields strong empirical gains and offers promising directions for domains beyond web navigation.

## 2 Related Work

**Scaffolded foundation models as web agents.** Prior works use external control structures to scaffold foundation models via modular prompting [31, 6, 32–37], programs [38**?** –41], or feedback

mechanisms [42–45]. These methods often rely on proprietary models like GPT-4 [46] or Claude [47]. Thus, progress is driven by designing better prompts and workflows for planning [48–51], self-correction [52], self-evaluation [53, 54], or by integrating external modules such as memory [55] or retrieval systems [56]. More recently, developing specialized agents has become a promising direction. ScribeAgent [14] first uses real-world data to demonstrate that simple fine-tuning can outperform most prompt-based agents. Prior work also builds automated data curation workflows [57–59] and distillation methods [18]. Despite these efforts, scaffolding methods do not enable agents to improve through interaction, and rely on fixed wrappers that lack adaptability across diverse environments.

**RL training for foundation model agents.** RL-based approaches enable agents to autonomously improve through interaction. Prior work has explored DPO [60], actor-critic [20, 21, 61], or distributed sampling [62]. Pipelines like PAE [19] and Learn-By-Interact [63] support automatic task generation, exploration, and labeling. However, most of these approaches lack mechanisms for test-time exploration, limiting the agent's ability to adapt its behavior over long horizons, especially under partially observable conditions. Our work addresses this limitation by scaling test-time interaction as an independent dimension, allowing agents to refine behavior while acting. Curriculum-based RL has been applied in AutoWebGLM [64] and WebRL [23], where curricula are based on estimated task difficulty derived from the complexity of LLM-generated instructions. While our approach also induces a progression from simpler to more complex behaviors, it does so by gradually increasing the allowed interaction horizon rather than relying on explicit measures of task difficulty.

**Scaling test-time compute.** Increasing test-time compute via best-of-$n$ sampling [29], beam search [65, 66], or verifiers [67–69] has shown to improve performance in reasoning-heavy tasks. In non-interactive settings like math and competitive coding, recent methods train models to generate long CoT and scale reasoning internally [e.g., 70, 27, 71**?** ]. As for multi-turn interactive settings, most existing works simply integrate CoT prompting into the agent system to enhance per-step reasoning [e.g., 72, 51]. EXACT [73] scales up the search process for each action, GenRM-CoT [74] the number of verifiers, and Jin et al. [75] the number of agents. However, none of these efforts studies the benefits of scaling over the time horizon, where the agent can explore alternatives, backtrack, or gather more information before committing to certain actions. Our work extends this line of research by introducing test-time scaling of interaction. As we will show in our empirical results (Section 4.2), the benefits of scaling test-time interaction go beyond test-time scaling (or "reasoning") within a given time step, because each extra step of interaction with the environment provides new information to the policy, whereas thinking for longer simply reorganizes information that the agent already has.

## 3 Problem Setup

We consider solving a web task as a finite-horizon sequential decision-making process guided by a reward objective[2]. Formally, the environment implements a transition function that evolves over time and provides an observation $o_t$ at step $t$ reflecting the current task state. The agent policy $\pi$ is parameterized by a multi-modal foundation model that maps observation history $o_{1:t-1}$ and action history $a_{1:t-1}$ to the next action $a_t$. These histories allow the policy to represent rich, context-dependent behaviors enabled via interaction. We denote the environment horizon, or the maximum number of interaction steps allowed in the environment, as $h$. For each task, the actual interaction process ends when the agent issues a stop signal or reaches the step limit $h$. Let $h_{\text{stop}} \in (0, h]$ denote the actual number of steps taken. The agent receives a reward of 1 for task success, and 0 otherwise.

**Observation space design.** Following [14, 19, 2], we consider an observation space consisting of the task goal, the URL, and a structured representation of the current web page that includes both the accessibility tree of the web page and a screenshot augmented with a set-of-marks overlay [1], which assigns a unique identifier to each element that the agent can interact with. While the agent has access to all past observations in principle, doing so quickly exhausts the context window in practice, so we truncate observation history to the most recent three steps, similar to prior works [19]. However, the agent still has access to all of its past actions. We show in Appendix B an example observation.

**Action space parameterization.** We adopt a discrete action space with six actions: click, type, scroll, go back, search (e.g., Google or Bing), and stop the task with an answer. Following Yang et al. [33], we do not consider compound actions like goto[url], as complex action spaces can hinder performance. For a detailed definition of each action, see the agent prompt in Appendix C.1.

---

[2]While this work centers on web agents, we believe the insights should generalize to other agent problem domains, and we hope future work will extend these ideas beyond web agents and web navigation.

# 4 Scaling Test-Time Interaction: A New Dimension of Agent Scaling

Prior methods for LLM test-time compute scaling usually scale the number of thinking tokens at each step [76**? ?** ], but this does not enable an agent to engage in longer interactions with the environment to collect new information. In principle, scaling the maximum number of interaction steps should allow the agent to employ richer behavioral strategies such as re-attempts, backtracking, and recovery. We will now verify this hypothesis via controlled experiments on WebArena [2]. We will then build upon these insights to develop ***TTI***, which trains agents to optimize test-time interactions.

**Controlled experiment setup.** We choose WebArena [2] as our testbed, primarily because it enables reproducible interaction with diverse domains (OneStopShop, Reddit, GitLab, CMS, and Open-StreetMap) and is equipped with ground truth evaluators. We randomly sample 62 tasks for testing and reserve the remaining 750 for online training (see Section 5.1). To ensure sufficient interaction, we set a generous test-time limit of $h = 30$, which is well above the average length of around 6 steps required by most tasks [55, 77]. Note that experiments in this section are for analysis rather than benchmarking purposes. We will show more experimental results on multiple benchmarks in Section 6.

To study the effect of increasing $h$, we use a simple prompting-based agent with Gemma 3 12B [78] base model, which observes the web page and outputs an action via a *single* model call. It does not leverage any retrieval, verifiers, or other external modules, ensuring any performance gains come solely from increased $h$ but not auxiliary scaffolding. We also study whether it is beneficial to prompt the agent to generate a reasoning trace before acting (see

Table 1: *Results averaged over three runs.*

| Prompt | Task SR (%) |
| --- | --- |
| Action Only | 14.76 |
| CoT | 23.81 |

Appendix C.1 for the templates). As Table 1 shows, CoT prompting yields significantly higher task success rate (SR) than direct action generation, setting a baseline of 23.81% on the test tasks. We thus adopt a simple chain-of-thought (CoT) approach as the default prompting strategy.

## 4.1 Scaling Test-Time Interaction by Acting Longer

To study the impact of interaction scaling, we introduce a purely inference-time "check-again" mechanism: after the agent issues task completion, we explicitly prompt it to reconsider its decision by "*You just signaled task completion. Let's pause and think again...*" We can extend re-checking to multiple checks. Detailed prompts are in Appendix C.4.

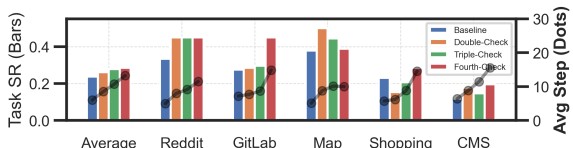

Figure 2: *Scaling test-time interaction by prompting the agent to "re-check" its answer.*

As shown in Figure 2, prompting the agent to re-check not only increases the interaction length $h_{stop}$ (line plots), but also improves the success rates on most domains (bar plots). When being asked to "check-again", the agent either reaffirms its decision (e.g., "*I previously stated the driving time was approximately 30 minutes....30 minutes seems plausible with typical traffic conditions. I'll stick with my previous answer.*") or revises it upon reflection (e.g., "*My apologies. Although the address book \*displays\* the desired address, the task requires me to \*change\* it....I should click button [24] to modify the address.*"). In particular, it changes its action ∼25% of the time after double-checking. This highlights the potential of interaction scaling: when given sufficient time, the agent is likely to explore alternatives before reaching an answer. The chances of the answer being correct could thus be higher.

However, we do observe that repeatedly prompting the agent to re-check can sometimes lead to confusion or hallucination, causing it to revise correct answers into incorrect ones. This may explain the performance drop observed in domains like Map. This is perhaps an inevitable limitation of scaling test-time interaction via *prompting* alone, akin to how prompting is not an effective way to even scale per-step test-time compute (see self-correction results for prompted models in **?** ]). We discuss this limitation further in Section 4.2 and address by *training* the agents explicitly.

## 4.2 Scaling Test-Time Interaction vs. Per-Step Test-Time Compute

Next, we examine the effect of scaling interaction compared to scaling per-step reasoning: Given a total token budget, should agents prioritize more interaction steps or generating longer reasoning traces at each step? To explore this, we study two conventional test-time compute scaling methods.

**Per-step budget forcing.** Following Muennighoff et al. [27], we prompt the agent to "wait and think again" after generating an initial CoT, encouraging more intermediate reasoning before it commits to an action. We vary the number of forced waits from 1 to 4. Despite trying various prompts (see Appendix C.4), our agent changes its actions only 15% of the time, and most of these changes involve reordering subtasks rather than exploring alternative paths or error correction.

**Per-step best-of-$n$.** At each step, we sample $n \in \{3, 5, 7\}$ candidate actions and select one via majority voting, similar to [14]. We did not scale $n$ up further because best-of-$n$ is compute-intensive for multi-step settings ($n \cdot h$ more expensive than the baseline per rollout), and we observe diminishing returns from sampling more actions, despite tuning sampling hyperparameters such as the temperature.

**Finding 1: Gaining new information via interaction beats thinking more a single step.** Figure 3 (top) plots the task success against total compute, measured by the number of tokens per trajectory in log scale. Among the three strategies, interaction scaling (green) shows the steepest upward trend, achieving the highest success rate as the allowed token budgets increase. Budget forcing (blue) yields moderate gains but plateaus around 0.26. Despite incurring the highest cost, best-of-$n$ (orange) brings the least improvements, suggesting that repeatedly sampling actions per step is a less effective use of compute in interactive tasks.

A natural question that arises here is: *how should we distribute a bounded compute budget between running more interaction steps vs. reasoning longer?* These two dimensions present different costs per unit, which may not be known apriori. Hence, we illustrate how scaling the number of interaction steps and the number of tokens per step affect performance and the total number of tokens individually. Figure 3 (bottom) decomposes total compute into tokens per step (y-axis) and steps per rollout (x-axis), so the shaded areas indicate the aver-

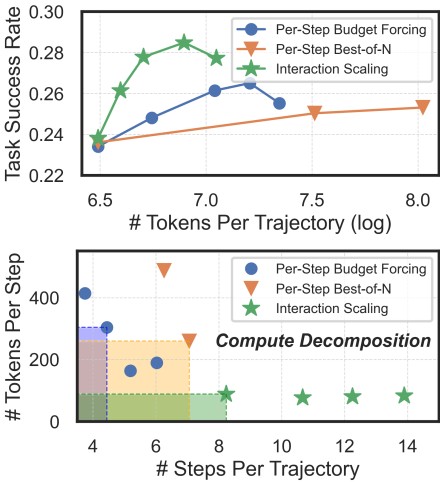

Figure 3: ***Top: Task success rate vs. total compute cost*** measured in tokens per trajectory, log scale. ***Bottom: Decompose compute*** into tokens per step and number of interaction steps (excluding the initial point for each approach).

age total compute per trajectory. Interaction scaling extends along the x-axis, while per-step reasoning scales along y-axis. We find that scaling *across* steps is more effective than scaling *within* steps in WebArena (Figure 3, top), likely because the former enables the agent to gather new information and enrich its context. This ability to query and observe external feedback is unique to agentic settings but not single-turn question-answering tasks. While standard per-step reasoning is constrained by the information already available at each step, our approach takes advantage of this dynamic interaction.

**Finding 2: Prompting alone is insufficient for interaction scaling.** While our results highlight the potential of scaling interaction, the "check-again" strategy only allows the agent to revisit its behavior upon task completion, it does not enable it to implement nuanced behaviors such as switching between exploration and exploitation in the middle of a rollout. We also experimented with combining interaction scaling with budget forcing and best-of-$n$ (Appendix Table 6) and observe that simply increasing test-time compute via prompting does not yield additive gains. In fact, the final downward trend in Figure 3 (top) suggests that asking the agent to re-check too many times or think for too long can confuse it and degrade performance. This shows the need for methods that *train* agents to optimize for best behavior when scaling test-time interaction, rather than naïve prompting.

## 5   *TTI*: Curriculum-Based Online RL for Scaling Interaction

How can we extend beyond prompting-based scaling to training agents that can effectively utilize interaction scaling? A natural starting point is to draw inspiration from current approaches for optimizing test-time compute [30, 71, 70] and extend these ideas to interactive settings. Specifically, we can run reinforcement learning (RL) with binary task rewards and longer task horizons. However, is this approach sufficient? We first describe the key challenges in learning to scale test-time interaction, and then develop our approach to address them via curriculum learning.

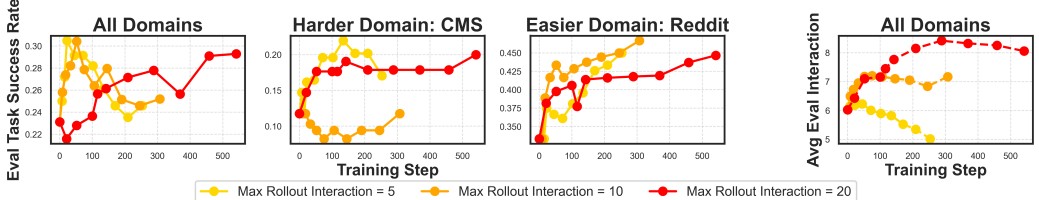

Figure 4: *Online RL with different values of maximum interaction horizon.* **Left:** success rates for different domains. "Harder" means lower success rate. **Right:** average rollout length ($h_{\text{stop}}$) on evaluation tasks.

## 5.1 Challenges of Training Agents with Fixed, Long Horizons

A natural way to encourage the agent to interact is to train at long horizons. To study this, we run the simplest form of REINFORCE [79] with binary rewards $R(\cdot)$, also known as online filtered behavior cloning (BC) or online STaR [20? ]. Only successful trajectories are retained, and the agent is updated by maximizing the log-likelihood of actions conditioned on those high-reward rollouts:

$$
\arg\max_{\theta} \quad \mathbb{E}_{\mathcal{T}\sim\text{tasks}} \left[ \mathbb{E}_{o_{0:h}, a_{0:h-1}\sim\pi(\cdot|\mathcal{T})} \left[ \underbrace{\left( \sum_{t=0}^{h-1} \log \pi_\theta(a_t \mid o_{\leq t}, \mathcal{T}) \right)}_{\text{likelihood of trajectory}} \cdot \mathbb{1} \left[ \underbrace{R(o_{0:h}, \mathcal{T}) = 1}_{\text{reward}} \right] \right] \right]
$$

$$(1)$$

We use filtered BC as it is stable throughout training (no negative gradient [? ]), has been utilized previously for training agents [20], and is a good "first-choice" algorithm for studying the role of interaction scaling. We scale the agent's horizon on WebArena, varying $h \in \{5, 10, 20\}$. Smaller $h$ exposes the agent only to exploitative rollouts that succeed within allowed time steps, while larger $h$ also includes more exploratory rollouts. We use the non-test tasks for rollout. Details are in Appendix D.

As shown in Figure 4 (left), the agent trained with $h = 5$ learns quickly, likely because on-policy RL is more sample-efficient at smaller horizons, but it also quickly overfits, and performance decreases with more training (x-axis) This agent often terminates prematurely during evaluation despite being allowed to interact for much longer time. Conversely, agents trained at longer horizons generally learn policies that are quite stochastic and learn significantly more slowly due to higher variance of the loss and credit assignment challenges due to longer horizons [e.g., 80–82]. We manually inspect the trajectories and find that the $h = 20$ agent tends to associate exploratory actions such as "going back" or "trying random links" with high rewards initially. Noisy credit assignment with $h = 20$ slows learning, and only after several iterations do the agents begin to recover and produce more robust policies. The impact of horizon is domain-dependent: in complex domains requiring exploration like CMS, long-horizon agents outperform, while in simpler settings like Reddit, performance differences are minimal. We also note that the total number of tokens generated per *action* slightly decreases throughout training, so the training does not incentivize increasing the length of reasoning directly.

Importantly, although the interaction length increases as expected for $h = 20$ (Figure 4, right), noisy credit assignment and slower learning suggests that simply setting $h$ to be large is insufficient to learn to scale test-time interaction reliably. These observations motivate our core idea: rather than fixing the horizon throughout training, our approach aims to scale their interaction length dynamically.

## 5.2 Our Approach: Curriculum Over Interaction Horizon

To address these challenges, we propose *TTI* (Test-Time Interaction), a curriculum-based online RL approach that trains the agent with short trajectories initially and gradually exposes it to longer ones. Existing curriculum learning methods in RL [e.g., 83–87] or web agents [23, 64] typically prioritize easier tasks before harder ones, often relying on predefined heuristics or external measures of task complexity. In contrast, we define curriculum progression in terms of the maximum number of steps an agent is allowed per trajectory. While interaction length can serve as a rough proxy for task difficulty, our approach does not require explicit labeling or estimation of task complexity.

***How do we design a curriculum over the interaction horizon*** $h$***?*** Ideally, the curriculum should allow the agent to first learn basic, "atomic" skills to solve easier tasks, then progressively tackle complex

ones via skill chaining and exploration. To enable this, we explored two strategies: **(1)** a conservative, additive increase in $h$ per iteration, giving the agent sufficient time to solidify core task-solving skills; and **(2)** a more aggressive, multiplicative increase, which assumes the agent can quickly acquire the basic skills and benefit from earlier exposure to exploratory behavior. Formally, for iteration $i$:

$$h_i := \text{clip}(h_{\min} + i, \ h_{\max}) \quad \text{(Additive curriculum)} \tag{2}$$

$$h_i := \text{clip}(h_{\min} \cdot i, \ h_{\max}) \quad \text{(Multiplicative curriculum)} \tag{3}$$

We store the rollouts in a replay buffer and assign higher weights to more recent trajectories. The full pseudocode for *TTI* and implementation details are provided in Appendix E.

**Empirical insights.** We instantiate these two strategies in WebArena, using the non-test tasks for online training. We set $h_{\min}$ to 10 and $h_{\max}$ to 30, and apply the schedules on top of filtered BC. Evaluation results after 10 iterations are shown in Table 2. Multiplicative curriculum outperforms the additive one, possibly because it exposes the agent to longer horizons early on and helps prevent it from overfitting prematurely to shortcut behaviors like always taking the shortest path. Based on these findings, we adopt the multiplicative curriculum as the default for *TTI*. Table 2 further shows that even with limited data ($\sim$700 training tasks), *TTI* outperforms fixed $h = 20$

Table 2: *Comparing various curricula.* A multiplicative curriculum produces the best success rate.

| Schedule | Task SR (%) |
|---|---|
| Additive | 29.50 |
| Multiplicative | 32.25 |

in Figure 4 by nearly 3%, using 40% fewer training steps over 10 iterations. Next, we demonstrate that this advantage carries over to large-scale online RL training.

## 6 Experiments: Scaling Up to Realistic Benchmarks

We now provide a comprehensive evaluation of *TTI* in large-scale, realistic environments. More generally, we demonstrate the effectiveness of training models to make good use of test-time interaction.

**Synthetic task generation and evaluation.** To enable large-scale training without training on the benchmark itself, we adopt synthetic task generation inspired by PAE [19]. Apart from prompting LLMs with seed examples or demos, we also deploy an exploratory agent to freely interact with websites and propose more grounded, diverse tasks. For evaluation, we leverage a prompting-based verifier that uses action histories and screenshots to label rollouts, achieving 88.9% accuracy compared to WebArena's ground-truth evaluator. Details and prompt templates are in Appendix F.1. We evaluate agents on (1) WebVoyager [1] with 427 tasks across 13 domains (we replace Google search with Bing due to ReCaptcha issues); and (2) full WebArena [2] with 812 tasks.

**Training.** We obtain 128K synthetic tasks across diverse real-world domains for WebVoyager and 11K tasks for WebArena's self-hosted domains (data released with the code). We train separate agents for each benchmark to avoid contaminating real-domain agents with synthetic environment data, as WebArena domains still have many differences from their real-world counterparts. We use Gemma 3 12B [78] as the base model, sampling 512 tasks per iteration for rollouts and updating with 512 successful on-policy trajectories. We apply a multiplicative curriculum with $h_{\min} = 10$ and $h_{\max} = 30$. We use vLLM [88] to sample rollouts and use DeepSpeed Zero 3 [89] with NVIDIA H100 GPUs for training. The evaluator is a Gemma 3 27B model, prompted to detect successful trajectories, which can then be used for the online filtered BC procedure. Other hyperparameters such as the number of iterations, learning rate, and the exact schedule of *TTI* can be found in Appendix F.3.

**Comparisons.** We evaluate zero-shot Gemma 3 12B and approaches that utilize a fixed horizon with $h \in \{10, 30\}$. We also compare to closed-source agents (e.g., those based on GPT-4 [46] and Claude [47]), open-weight models trained on proprietary data (e.g., UI-TARS [90]), and fully open-weight, open-data models (e.g., PAE [19]).

### 6.1 WebVoyager Results and Analysis

***State-of-the-art open-weight, open-data performance.*** We report the overall task success rates (SR) on WebVoyager in Table 3 and Figure **??** (left). The *TTI*-trained Gemma 3 12B achieves an average SR of 64.8%, setting a new state-of-the-art among open agents trained purely on public data. While previous methods such as UI-TARS achieves a strong SR of 84.8%, they rely on private human-annotated data that remains inaccessible to the open-source community. In contrast, *TTI* is trained entirely on synthetic data and interestingly, this data is generated by the base model (Gemma

Table 3: **WebVoyager results. TTI** attains the best performance among open-weight agents trained on open-source synthetic data. Baseline results are taken from Zhou et al. [19], Qin et al. [90].

| Model | Average | Allrecipes | Amazon | Apple | ArXiv | GitHub | ESPN | Coursera | Cambridge | BBC | Map | Search | HuggingFace | WolframAlpha |
|---|---|---|---|---|---|---|---|---|---|---|---|---|---|---|
| **Proprietary Model** | | | | | | | | | | | | | | |
| Claude 3.7 | 84.1 | - | - | - | - | - | - | - | - | - | - | - | - | - |
| Claude 3.5 | 50.5 | 50.0 | 68.3 | 60.4 | 46.5 | 58.5 | 27.3 | 78.6 | 86.0 | 36.6 | 58.5 | 30.2 | 44.2 | 66.7 |
| OpenAI CUA | 87.0 | - | - | - | - | - | - | - | - | - | - | - | - | - |
| Agent E | 73.1 | 71.1 | 70.7 | 74.4 | 62.8 | 82.9 | 77.3 | 85.7 | 81.4 | 73.8 | 87.8 | 90.7 | 81.0 | 95.7 |
| **Open Model, Proprietary Human-Annotated Data** | | | | | | | | | | | | | | |
| UI-TARS-1.5 | 84.8 | - | - | - | - | - | - | - | - | - | - | - | - | - |
| **Open Model, Open Synthetic Data** | | | | | | | | | | | | | | |
| LLaVa-34B SFT | 22.2 | 6.8 | 26.8 | 23.3 | 16.3 | 4.9 | 8.6 | 26.8 | 67.4 | 16.7 | 12.2 | 23.3 | 20.9 | 38.1 |
| PAE-LLaVa-7B | 22.3 | 14.3 | 37.5 | 17.5 | 19.0 | 14.6 | 0.0 | 33.3 | 52.4 | 18.6 | 22.5 | 23.3 | 19.0 | 24.4 |
| PAE-LLaVa-34B | 33.0 | 22.7 | 53.7 | 38.5 | 25.6 | 14.6 | 13.6 | 42.9 | 74.4 | 39.0 | 22.0 | 18.6 | 25.6 | 42.9 |
| Gemma 3 12B | 55.8 | 25.7 | 32.3 | 45.5 | 60.6 | 54.8 | **60.6** | 56.3 | 69.6 | 65.6 | 54.8 | **72.7** | 66.7 | 61.1 |
| Fixed $h = 10$ | 59.1 | 25.7 | **74.1** | 51.5 | **75.7** | **70.9** | 44.1 | **59.3** | 66.7 | 50.0 | 41.9 | 60.6 | 75.5 | 72.2 |
| Fixed $h = 30$ | 45.2 | 20.0 | 41.9 | 60.6 | 42.4 | 41.9 | 50.0 | 34.4 | 60.6 | 25.0 | 29.0 | 63.6 | 45.5 | 69.4 |
| *TTI* (Ours) | **64.8** | **57.1** | 48.3 | **69.6** | 66.6 | 45.2 | 56.3 | 46.9 | **85.2** | **81.2** | **66.7** | **72.7** | **75.7** | **79.4** |

3 12B) itself, meaning that our training protocol implements a form of *self-improvement*. **TTI** also obtains the highest SR in 8 out of 13 domains, illustrating its efficacy.

**TTI** *outperforms fixed-horizon via adaptive exploration.* Table 3 also shows that our curriculum approach outperforms fixed $h = 10$ baseline by 5.7% and fixed $h = 30$ baseline by 19.6% in average accuracy. To better understand the use of interaction within a training rollout, we plot the average number of interaction steps on a held-out validation set with 78 tasks in Figure 5 (a). Note that the agent trained with $h = 10$ learns to continuously reduce the maximum number of steps it spends in a rollout, while $h = 30$ quickly drifts into aimless exploration and executes a larger number of steps pre-maturely in training, hindering performance. This aligns with our findings in Section 5.1 that simply running training at a longer horizon may not be sufficient for obtaining effective interaction scaling. In fact, we find that when training with **TTI**, the interaction length of the agent's rollouts first decreases but then starts to increase as the maximum allowed horizon increases, indicating that an adaptive curriculum enables effective interaction scaling.

Figure 5 (d) shows that the task success rate also grows over time and correlates with the expanding horizon. While the average task success rates for **TTI** are better, we observe notable per-domain differences. Figure 5 (e) shows representative per-domain success rates. On domains like Allrecipes and Cambridge, **TTI** significantly outperforms fixed-horizon and zero-shot approaches, improving success rates by 31.4% and 15.6%, respectively, likely because these domains are highly information-dense and benefit from extended exploration enabled by adaptive interaction scaling. However, in domains like Amazon and GitHub, **TTI** underperforms the baselines. We notice that the base model already has strong knowledge about domain-specific terminologies (e.g., commit history, forks, stars) in these domains, resulting in high base performance. Inspecting the rollouts, we find that instead of using built-in filters and sorting, **TTI** can engage in exploration behaviors such as initiating Bing searches or consulting external sites. This exposes the agent to noisy or distracting information, reducing task success. We discuss these cases in Section 6.2.

*Learning dynamics of* **TTI**. To study how **TTI** enhances the "within-rollout" exploration capabilities of the agent, we measure the number of *GoBack* and *Bing* actions over the course of training. *GoBack* actions measure the number of retries the agent makes within an episode to get unstuck during exploration. *Bing* actions correspond to the number of times the agent attempts to seek information by moving to `bing.com`. As shown in Figure 5 (a, b, and d), the performance of **TTI** improves substantially as the number of GoBack and Bing actions and the trajectory length grow.

Also note that the trajectory length and the numbers of *GoBack* and *Bing* actions begin to increase with **TTI**, once the maximum allowed horizon length is increased as a part of the curriculum schedule (this regime is shown by the green shaded area in Figure 5). In contrast, these quantities continuously decrease over the course of training for the run with a lower number of maximum interaction steps ($h = 10$). We also find that the trajectory length shoots up substantially for the run with $h = 30$ and this correlates with worse performance. Finally, as shown in Figure 5 (c), as the agent's trajectory grows longer with **TTI**, the number of tokens appearing in per-step reasoning actually becomes smaller. This implies that our agent is automatically learning to tradeoff interaction for per-step compute in order to attain higher performance, and perhaps prevents any issues with overthinking.

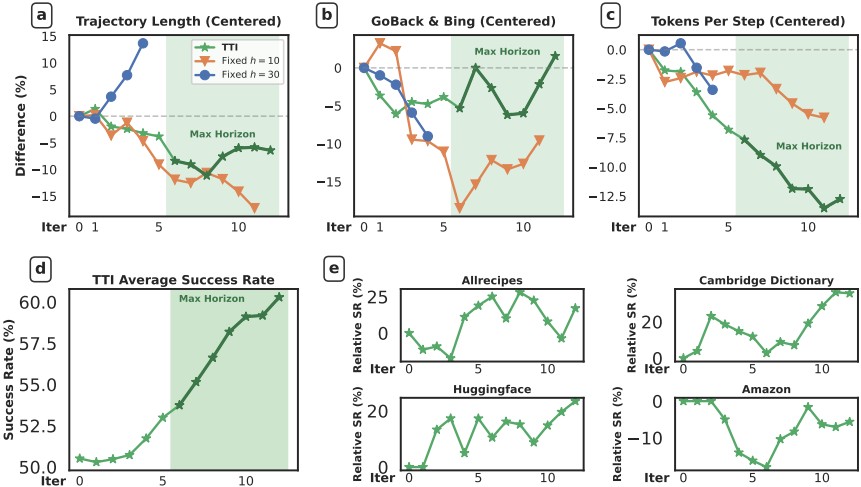

Figure 5: *Dynamics of* **TTI** *during training.* The green area represents the phase where the maximum allowed interaction horizon is the largest ($h = 30$), per our multiplicative schedule. All results are evaluated on a held-out subset of WebVoyager, not on the training tasks. **a:** Average trajectory length, i.e. the average number of steps taken in a trajectory normalized by the average length at the first iteration (iteration 0). **b:** Ratio of the sum of GoBack and Bing actions out of all actions normalized by the first iteration. **c:** The average number of tokens (CoT lengths) per action. **d:** Average task success rates for **TTI**. **e:** Per-domain success rates for **TTI**.

## 6.2 Case Studies: Strengths and Failure Modes

We conduct detailed case studies to analyze how **TTI** behaves across tasks and domains. These cases highlight both the strengths and remaining limitations of our approach.

**Strength: Effective exploration in complex tasks (example visualized in Appendix F.4).** For complex, exploratory tasks that require information retrieval, **TTI** trains the agent to extend its interaction horizon through search and backtracking, thus gathering and comparing information before making decisions. For instance, when tasked to find the baking temperature of an apple pie recipe with 4+ stars and 50+ reviews, our agent first selects a recipe but encounters a pop-up it cannot dismiss due to backend issues. It then tries another recipe but finds no baking temperature. Returning to the listing again, it correctly identifies one that meets all criteria. We also observe that such behaviors emerge progressively. In early training with shorter horizon, **TTI**-agent tends to stick to the first recipe it finds, keeps retrying it and saying *"I remember seeing one with 613 ratings earlier"* instead of seeking alternatives. Only after the training horizon becomes longer does it learn to explore and backtrack. This illustrates that when **TTI** runs a curriculum over interaction length, it teaches agents to adjust their horizon *within* a task and shift from exploitation to exploration. In contrast, training with a fixed short horizon can make it difficult to develop such exploratory behaviors.

**Strength: Strategic exploitation in simple tasks (Appendix F.5).** For simpler tasks with clear, deterministic paths (e.g., form filling or direct lookups), **TTI**-agent completes tasks efficiently without over-exploration. For example, when instructed to find the "top trending open-source project on machine learning" in GitHub, the agent goes directly to the Open-Source menu, selects the Trending tab, and performs search. This shows that **TTI** balances exploration and exploitation based on task context.

Despite these strengths, we also observe characteristic failure modes that point to areas for improvement and may partly explain the agent's lower performance on domains like GitHub.

**Failure mode: over-reliance on resets (Appendix F.6).** When an action fails, our agent can reset the task by returning to the Bing search page rather than attempting domain-specific recovery actions (e.g., revisiting menus, refining filters). This suggests the agent treats search as a universal fallback. We also observe repeated resets within the same trajectory, indicating a lack of adaptive error recovery. While agents can extend horizons through both resetting and backtracking, the latter is often more natural. This highlights an area where **TTI** could improve by guiding exploration more systematically and enforcing structure, e.g., by using dense rewards [71, 68].

**Failure mode: limited self-verification (Appendix F.7).** We also observe that the agent can fail to verify its actions against the task goal. In one case, the agent identifies a 2021 GitHub repository for a task requiring one from 2022. While it explicitly acknowledges the mismatch, *"It was created in*

Table 4: *WebArena results.* For proprietary agents, we include the top 8 from the official leaderboard. Observe that TTI attains the best performance on an average, with especially better performance on the CMS domain.

| | Method | Backbone | Average | Shopping | CMS | Reddit | GitLab | Maps |
|---|---|---|---|---|---|---|---|---|
| **Proprietary-Based** | IBM CUGA [92] | - | 61.7 | - | - | - | - | - |
| | OpenAI CUA [4] | - | 58.1 | - | - | - | - | - |
| | Jace AI [93] | - | 57.1 | - | - | - | - | - |
| | ScribeAgent [14] | GPT-4o + Qwen2.5 32B | 53.0 | 45.8 | 37.9 | 73.7 | 59.7 | 56.3 |
| | AgentSymbiotic [18] | Claude 3.5 + Llama 3.1 8B | 48.5 | 48.7 | 41.2 | 63.2 | 47.2 | 57.8 |
| | Learn-by-Interact [63] | Claude 3.5 Sonnet | 48 | - | - | - | - | - |
| | AgentOccam-Judge [33] | GPT-4 | 45.7 | 43.3 | 46.2 | 67.0 | 38.9 | 52.3 |
| | WebPilot [34] | GPT-4o | 37.2 | 36.9 | 24.7 | 65.1 | 39.4 | 33.9 |
| **Fully Open-Source** | Learn-by-Interact [63] | Codestral 22B | 24.2 | - | - | - | - | - |
| **(Self-Improvement)** | AgentTrek [37] | Qwen2.5 32B | 22.4 | - | - | - | - | - |
| | AutoWebGLM [64] | ChatGLM3 6B | 18.2 | - | - | - | - | - |
| | NNetnav [94] | Llama 3.1 8B | 7.2 | 7.4 | 4.2 | 0 | 0 | 28.5 |
| | Zero-Shot Baseline | Gemma 3 12B | 18.3 | 26.7 | 8.7 | 30.9 | 5.5 | 27.7 |
| | Fixed $h = 10$ | Gemma 3 12B | 23.8 | 28.4 | **15.6** | 26.0 | 13.2 | 34.7 |
| | Fixed $h = 30$ | Gemma 3 12B | 19.0 | 25.7 | 9.7 | 29.8 | 8.7 | 28.57 |
| | *TTI* (Ours) | Gemma 3 12B | **26.1** | **33.9** | 15.5 | **35.3** | **15.7** | **40.5** |

*2021, not 2022, so it doesn't meet the criteria"*, it still submits it as the answer. This implies limited self-verification ability and could be mitigated by longer, more deliberate per-step reasoning. An important next step is to enrich *TTI* with the cognitive behaviors that enable per-step reasoning [91].

### 6.3 WebArena Results and Analysis

We further assess *TTI* on the full WebArena [2] (we only use the prompting-based verifier for training, but use the original benchmark evaluators for evaluation). As shown in Table 4 and Figure **??** (right), *TTI* obtains the highest performance among open-source agents trained entirely using a self-improvement procedure on self-generated data, without relying on proprietary models for task completion or distillation. While *TTI* improves over the zero-shot baseline by 7.8%, the gains are smaller than on WebVoyager, possibly because: (1) WebArena tasks are more complex, as reflected in lower accuracies even for proprietary models, leading to fewer successful rollouts per iteration and slower learning; (2) Agents occasionally attempt actions that are valid in real-world domains but invalid in WebArena counterparts (for example, search works reliably on Reddit but fails in WebArena's Postmill due to environment bugs). More experiment details are in Appendix G.

**Further scaling.** While *TTI* equips agents with the ability to adjust their interaction horizon during deployment, an open question remains: Can we further amplify performance by combining TTI with inference-time interaction scaling techniques such as re-checking as discussed in Section 4? To explore this, we apply the "check-again" strategy (Section 4) to intermediate *TTI* checkpoints. Due to the high evaluation cost associated with evaluating on full WebArena, we leverage the WebArena subset checkpoints obtained in Section 5.2. As shown in Figure 6, applying re-checking on top of *TTI* improves task success

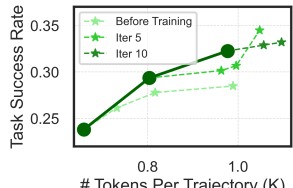

Figure 6: *We apply test-time re-checks to* TTI *checkpoints.*

across various training stages. The benefits are more obvious in the early stage, when the agent has a stronger bias to terminate prematurely. As training progresses, *TTI* encourages longer interaction that naturally incorporate behaviors like re-checking, reducing the benefit of explicit re-checks.

## 7 Conclusion

In this work, we introduced interaction scaling as a new dimension of test-time scaling for interactive agents. Through empirical studies on web agents, we validate that interaction scaling enables agents to explore and adapt dynamically, significantly improving task performance. We hope that this work opens new directions in agentic reasoning and inspires broader applications beyond web navigation.

**Limitations and future work.** Our experiments are limited to web environments; extending this method to other domains like robotics or open-world games requires further exploration. Besides, scaling interaction steps increases computational costs during both inference and training. Although our adaptive scheduling helps, more efficient handling of long interactions is needed. In addition, our training relies on simple behavior cloning; future work could incorporate more advanced RL methods like PPO [95], GRPO [70] to improve performance. Lastly, due to high compute cost, we only ran the full benchmark once per setting, limiting the ability to quantify variance from policy, environment, and evaluator stochasticity. Future work should explore multiple runs or more robust evaluation.

## Acknowledgments

We thank Jiayi Pan, Shanda Li, and Yuxiao Qu for feedback and informative discussions on an earlier version of this paper. This work was supported in part by the National Science Foundation grants IIS1705121, IIS1838017, IIS2046613, IIS2112471, the Office of Naval Research under N00014-24-1-2206, and funding from Meta, Morgan Stanley, Amazon, Google, and Scribe. We specially thank Scribe, The AGI company, and CMU FLAME Center (Orchard cluster) for providing a part of the GPU resources that supported this work. Any opinions, findings and recommendations expressed in this material are those of the author(s) and do not necessarily reflect the views of any of these funding agencies or employers.

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

## A  Broader Impact

This work contributes to the development of more adaptive and capable AI agents by introducing a new test-time scaling dimension focused on interaction rather than per-step reasoning alone. While this approach improves robustness and generalization in open-ended environments, it also raises important considerations. Increased agent autonomy can amplify both the benefits and risks of deployment in real-world systems. Moreover, agents capable of richer behaviors could be applied to sensitive domains (e.g., customer service, education, or automation workflows) where unintended actions could have large impacts. We encourage future work to consider ethical safeguards, interpretability tools, and human-in-the-loop designs when deploying interaction-scaled agents. Our experiments are conducted entirely in simulated environments, and we hope this work inspires further research on controllable and trustworthy agent behavior under realistic constraints.

## B  Observation Space Design

We use the screenshot accompanied with the web page's accessibility tree as our main observation. We study two versions of accessibility tree. **Rich accessibility tree** is modified from the WebArena code and looks like:

[21]: RootWebArea 'Dashboard / Magento Admin' focused: True; [0]: link 'Magento Admin Panel'; [1]: link 'DASHBOARD'; [2]: link 'SALES'; [3]: link 'CATALOG'; [4]: link 'CUSTOMERS'; [5]: link 'MARKETING'; [6]: link 'CONTENT'; [7]: link 'REPORTS'; [8]: link 'STORES'; [22]: link 'SYSTEM'; [23]: link 'FIND PARTNERS & EXTENSIONS'; [24]: heading 'Dashboard'; [9]: link 'admin'; [10]: link ''; [25]: StaticText 'Scope:'; [12]: button 'All Store Views' hasPopup: menu; [13]: link 'What is this?'; [14]: button 'Reload Data'...

**Simple accessibility tree** is modified from the PAE code and looks like:

[1]: "Dashboard"; [2]: "Sales"; [3]: "Catalog"; [4]: "Customers"; [5]: "Marketing"; [6]: "Content"; [7]: "Reports"; [8]: "Stores"; [9]: "admin"; [12]: <button> "All Store Views"; [13]: "What is this?"; [14]: <button> "Reload Data"; [15]: "Go to Advanced Reporting"; [16]: "here";...

Rich tree contains more details such as the HTML tag and attributes like `required`, `hasPopup` compared to simple tree. However, it is much longer than simple tree and hence harder to optimize due to the increased context length. As simple tree gives more steady training dynamics, we mainly use it for our experiments.

## C  Preliminary Test-Time Experiments on WebArena

### C.1  General Prompts

---
**General Prompt**

Imagine you are an agent browsing the web, just like humans. Now you need to complete a task. In each iteration, you will receive an observation that includes the accessibility tree of the webpage and a screenshot of the current viewpoint. The accessbility tree contains information about the web elements and their properties. The screenshot will feature numerical labels placed in the TOP LEFT corner of web elements in th current viewpoint. Carefully analyze the webpage information to identify the numerical label corresponding to the web element that requires interaction, then follow the guidelines and choose one of the following actions:
1. Click a web element.
2. Delete existing content in a textbox and then type content.
3. Scroll up or down the whole window.
4. Go back, returning to the previous webpage.
5. Answer. This action should only be chosen when all questions in the task have been solved.
Correspondingly, action should STRICTLY follow the format specified by one of the following lines:
Click [numerical_label]
Type [numerical_label] [content]

---

Scroll [up/down]
GoBack
ANSWER [content]
Some examples are:
Click [8]
Type [22] [Boston]
Scroll [down]
ANSWER [06516]
Key guidelines you MUST follow:
* Action guidelines *
- Use either screenshot or accessibility tree to obtain the numerical_label. Sometimes the accessibility tree captures more elements than the screenshot. It's safe to select these elements without scrolling
- For text input, use Type action directly (no need to click first). All existing texts in the textbox will be deleted automatically before typing
- Preserve text inside quotation marks exactly as provided by user
- You must not repeat the same actions over and over again. If the same action doesn't work, try alternative approaches
- Use ANSWER only after completing ALL task requirements
- Wrap content for Type and ANSWER with square brackets '[]'
- Do not add quotation marks for search queries
* Web navigation hints *
{hint}
Your reply should strictly follow the format:
Thought: Your reasoning trace. A good practice is to summarize information on the current web page that are relevant to the task goal, then generate a high-level plan that contains the sequence of actions you probably need to take
Action: Based on this reasoning, identify the single most optimal action. You should output it in the format specified above (under "STRICTLY follow the format")
After each action, you'll receive a new observation. Proceed until task completion. Now solve the following task.
Task: {task_goal}
Current URL: {url}
Screenshot of current viewpoint: attached
Accessibility tree of current viewpoint: {accessibility_tree}

Beyond the above CoT prompt, we also tried using a more complex prompt for the thought process. However, this does not lead to significant gain in downstream accuracy (see Table 5), but it could increase training and inference cost, so we did not use it in the end.

**Complex Prompt**

Thought: You must analyze the current webpage thoroughly to guide your decision-making. Show your reasoning through these steps:
- Summarization: Begin by understanding the page context - identify what type of page you're on (search results, form, article, etc.) and how it relates to your objective. Summarize important information on the webpage that might be relevant to task completion. Especially when the task requires to return some answers to a specific question, you should note down intermediate information that helps generate the answer.
- Planning: Generate a checklist of subtasks required for completion and cross-out the subtasks you've completed. Identify the next logical subtask.
- Verification: Verify all information you've entered so far. Check that your inputs match requirements in terms of spelling and format (you should not change the user-specified information, even if there're grammar errors). Verify if any selections for dropdown items align with the task objective. Identify if there're necessary fields that have not been filled in. Note that if the last few steps are repeating the same action, there must be missing or incorrect information.

- Backtracking: If the task requires exploring multiple webpages (e.g., orders, posts, item pages, etc) to find out an answer, consider if you need to issue GoBack and return to the previous web page.
- Candidate Generation: After all the above reasoning, list the most relevant possible actions, evaluate pros and cons of each action, and finally select the most effective action to progress task.
  Action: Choose ONE of the following action formats:
- Click [numerical_label] - Click a specific element
- Type [numerical_label] [content] - Input text into a field
- Scroll [up/down] - Navigate the page vertically
- GoBack - Return to previous webpage
- ANSWER [content] - Provide final answer when task is complete

## C.2 WebArena Prompts

Below are the content replacing "{hint}" in the general prompt.

### General Hint

- Always save progress through appropriate buttons (Save, Submit, Post, etc.)
- Always remember to interact with dropdown options after expanding
- Clear filters before setting new ones

### Reddit

- Always save progress through appropriate buttons (Save, Submit, Post, etc.)
- Always remember to interact with dropdown options after expanding
- Pay attention to words like "latest", "newest", "hottest" in the task objective, which require clicking the dropdown menu and select "New" or "Top" with the correct time range
- When selecting a subforum, you can either browse the dropdown menu in the "Submit" page or navigate to "Forums" and check all subforums by clicking on "Next" to go over all pages. You must try to find a subforum that exactly matches your query. If there's no exact match, pick the most relevant one, ideally the subforum is about objects or locations contained in the given objective
- "Trending" means "hot"
- To find out all posts or replies from a user, click the user name and then click "Submissions" or "Comments"

### CMS

- Always save progress through appropriate buttons (Save, Submit, Post, etc.)
- Always remember to interact with dropdown options after expanding
- Clear filters before setting new ones
- Use date format: month/day/year (e.g., 1/1/16, 12/31/24)
- When searching phone numbers, remove the country code
- When searching product name, use single but not plural form
- When the web page contains a table, aggregate the rows with the same item

### Shopping

- Always save progress through appropriate buttons (Save, Submit, Post, etc.)
- Always remember to interact with dropdown options after expanding
- Sort items by price by clicking the dropdown menu and set descending/ascending direction
- When searching product name, use single but not plural form
- If the objective requires only finding an item, stop at the item page without adding to cart
- To find out the quality of a product, search the item, click on review, and inspect its review

> - Click "Page Next" to iterate over all orders
> - Since there's no way to filter order history, click "View Order" for every order within a date range and inspect individually. If the condition is not met, go back

---

**GitLab**

- Always save progress through appropriate buttons (Save, Submit, Post, etc.)
- Always remember to interact with dropdown options after expanding
- Clear filters before setting new ones
- When searching a repo in gitlab, type only the project name after "/" in the search box

---

**Map**

- Always remember to interact with dropdown options after expanding
- When searching for a place, remove prepositions like in/on/by/at. For example, use "starbucks, craig street" instead of "starbucks on craig street". Put the city name at the end
- When there is no results shown up after search, rephrase the address and try again
- To find direction between two points, after entering the from and to addresses, select the correct transportation (foot/bicycle/car) before clicking "Go"
- When the given location is not a geological address, use your knowledge to infer the address

## C.3 CoT Experiments for Base Agent

To enable efficient rollout collection, we spin up multiple Docker containers on a single GPU according to the official WebArena repository. We use the vLLM [88] engine for inference and apply the following inference hyperparameters for most of our experiments.

- max_new_tokens: 1024

- max_attached_imgs: 4

- temperature: 1

- top_p: 0.95

We randomly subsample 62 test tasks for analysis purposes. Below are the results of zero-shot agent vs CoT prompting. "CoT" uses the "General Prompt" in Section C.1. "Complex CoT" uses the "Complex Prompt" in Section C.1. .

Table 5: Base agent results averaged over 3 runs on WebArena subset.

| Prompt | Task SR (%) |
|---|---|
| Action Only | 14.76 |
| CoT | 23.81 |
| Complex CoT | 23.33 |

## C.4 Scaling Trade-off Experiments

**"Check-again" for interaction scaling.** After the agent outputs the task-stop signal, we append the following prompts to the observation to induce it to check again.

---

**Check-Again Prompt**

Important: You returned an answer in the last step. Let's pause, check the web page, and think again. If you still think the task is finished, double-check your answer, revise it if need, and return a final answer. If not, continue the task. Your output should still be in the same "Thought:...Action:..." format.

Table 6: Comparing different inference-time prompting strategies. Results averaged over 3 runs on WebArena subset. All methods are applied once.

| Inference-Time Strategy | Task SR (%) |
|---|---|
| Baseline | 23.81 |
| Check-again | 26.14 |
| Budget-forcing | 24.81 |
| Best-of-$n$ | 25.03 |
| Check-again + Budget-forcing | 26.33 |
| Check-again + Best-of-$n$ | 27.36 |

When applying multiple re-checks, we slightly vary the prompts such as '*"Before you finalize the answer, re-evaluate it in terms of the current web page—what exactly supports or contradicts it?"* or *"Why do I believe this answer is correct? What on the page justifies it? Could an alternative answer be better?"* Please refer to the code base for the exact prompt used.

**Per-step budget forcing.** Following [71], we use the phrases below to induce longer per-step thinking. The phrases are different to ensure that the model does not run into the scenario of endless repeating a phrase.

- First time: Wait, let me think deeper.
- Second time: But let me double-check.
- Third time: But hold on.

**Per-step best-of-$n$.** We tried both selecting by log likelihood and majority voting, with the latter showing slightly better results.

**Additional results for combined scaling.** Beyond evaluating each scaling method separately, we also tried combining methods along different axes.

# D  Online Filtered BC on WebArena

We use the following hyperparameters to obtain the training curves in Table 4. During training, the `vision_tower` of Gemma 3 is kept frozen because it is frozen during pretraining. Other hyperparameters can be found in our code in the supplementary material.

- num_iteration: 10
- actor_epochs: 1 # number of epochs to update the actor
- rollout_size: 512
- num_update_sample_per_iteration: 512
- lr: 1e-6
- optimizer: AdamW
- scheduler: WarmupCosineLR
- batch_size: 4
- grad_accum_steps: 2
- eval_horizon: 30

# E  *TTI* Implementation

We provide the pseudocode in Algorithm 1. For the replay buffer, to encourage the agent to learn from more recent examples, we assign weights based on recency when sampling rollouts to update the agent: for the $k$-th trajectory added to the buffer, its weight is $\frac{k}{|\mathcal{D}|}$.

**Algorithm 1 *TTI*: Filtered Behavior Cloning with Interaction Scheduling**

1: **Input:** Agent policy $\pi_\theta$, Evaluator $\mathcal{R}$, Environment $\mathcal{E}$, Learning rate $\alpha$, Replay buffer $\mathcal{D}$, Interaction scheduler hyperparameters $h_{\min}, h_{\max}$
2: Initialize policy $\pi_\theta$ from pretrained model
3: Initialize replay buffer $\mathcal{D} \leftarrow \{\}$
4: **for** each episode $i$ **do**
5:     Set interaction horizon $h_i \leftarrow \text{get\_schedule}(i, h_{\min}, h_{\max})$
6:     **for** each rollout to collect **do**
7:         Initialize environment: $s_0 \sim \mathcal{E}$
8:         **for** each $t$ in $[1, h_i]$ **do**
9:             Observe current state $s_t$
10:             Predict action $\hat{a}_t \leftarrow \pi_\theta(s_t)$
11:             Execute action $\hat{a}_t$ in environment
12:             Observe next state $s_{t+1}$
13:             **if** episode done **then**
14:                 Compute reward $r_t \leftarrow \mathcal{R}(s_t, \hat{a}_t)$
15:             **else**
16:                 $r_t \leftarrow 0$
17:             **end if**
18:             Store transition: $\mathcal{D} \leftarrow \mathcal{D} \cup \{(s_t, \hat{a}_t, r_t, s_{t+1})\}$
19:         **end for**
20:     **end for**
21:     **for** sample successful trajectory in $\mathcal{D}$ **do**
22:         **for** $t = 1$ to $h_{\text{stop}}$ **do**
23:             Accumulate loss: $L(\theta) \leftarrow L(\theta) + \text{CrossEntropy}(\pi_\theta(s_t), \hat{a}_t)$
24:         **end for**
25:     **end for**
26:     Update policy: $\theta \leftarrow \theta - \alpha \nabla_\theta L(\theta)$
27: **end for**

# F  WebVoyager Experiments

## F.1  Task Generator & Evaluator Prompt

---
**Task Generator Prompt**

You are a website exploration assistant tasked with discovering potential tasks on websites. These tasks should be similar to a user-specified task and aim to complete some high-level goals such as booking restaurants in a website. Your goal is to freely explore websites and propose tasks similar to a given set of examples. For each iteration, you'll receive:
- An observation with the webpage's accessibility tree
- A screenshot showing numerical labels in the TOP LEFT corner of web elements
You will then generate possible tasks while exploring the website. You should imagine tasks that are likely proposed by a most likely user of this website. You'll be given a set of examples for reference, but you must not output tasks that are the same as the given examples. The generated tasks must be realistic and at least require 3 steps to complete. It cannot be too simple.
## Response Format and Available Actions
Your reply for each iteration must strictly follow this format:
Thought: Analyze the current webpage thoroughly to guide your exploration. Examine the webpage's structure, content, and interactive elements to identify potential tasks that users might perform on this site. Decide whether you want to keep exploring or output some tasks
Tasks: If you think you are ready to generate some tasks, output them in the following format (note that different tasks are separated with double semicolons): GENERATE [task1;answer1;;task2;answer2]
Action: Then, to continue with your exploration, choose ONE of the following action formats:

---

- Click [numerical_label] - Click a specific element
- Type [numerical_label] [content] - Input text into a field
- Scroll [up/down] - Navigate the page vertically
- GoBack - Return to previous webpage
Examples:
Click [8]
Type [22] [Boston]
Scroll [down]
GENERATE [Find the company's phone number;(555) 123-4567;;Locate the price of the basic subscription plan;$19.99/month]
Your final output should look like:
Thought: ...
Tasks: GENERATE [...] (this is optional, only generate when you are confident)
Action: ...
## Critical Guidelines
### Action Rules
- Use either screenshot or accessibility tree to obtain the numerical_label
- For text input, use Type action directly (no need to click first)
- Ensure proposed tasks are diverse and demonstrate different aspects of the website. The tasks must have diverse difficulty and require different number of steps (3-20) to complete.
- Tasks should be clear, specific, achievable, and self-contained. It cannot be too general, e.g., related to äny post, äny product, äny place. It must not depend on any context or actions that you have performed, i.e., you must assume zero prior knowledge when someone wants to complete the task
- Your task should be objective and unambiguous. The carry-out of the task should NOT BE DEPENDENT on the user's personal information such as the CURRENT TIME OR LOCATION
- Your tasks should be able to be evaluated OBJECTIVELY. That is, by looking at the last three screenshots and the answer provided by an agent, it should be possible to tell without ambiguity whether the task was completed successfully or not
- Answers should be precise (e.g., exact prices, specific information, exact text)
- Your should output both operational tasks (the goal is to complete some steps) and information retrieval tasks (the goal is to find some answer to return)
- You must refer to the examples given and mimic the complexity and task structure. See how these tasks are self-contained and realistic
- Your proposed task cannot be a single action like click, type! Tasks like 'Determine the number of uses for that term' is unacceptable because it is ambiguous as a stand-alone task; 'Uncheck Use system value' is unacceptable because it is not a complete task; 'Locate the total revenue for the last month' is unacceptable because 'last month' is ambiguous;
After each action, you'll receive a new observation. Continue exploring and generating tasks.
Here're some examples: {example}
Current URL: {url}
Screenshot of current viewpoint: attached
Accessibility tree of current viewpoint: {accessibility_tree}

---

## Evaluator Prompt

You are an expert in evaluating the performance of a web navigation agent. The agent is designed to help a human user navigate a website to complete a task. Your goal is to decide whether the agent's execution is successful or not.
As an evaluator, you will be presented with three primary components to assist you in your role:
1. Web Task Instruction: This is a clear and specific directive provided in natural language, detailing the online activity to be carried out.
2. Result Response: This is a textual response obtained after the execution of the web task. It serves as textual result in response to the instruction.

3. Result Screenshots: This is a visual representation of the screen showing the result or intermediate state of performing a web task. It serves as visual proof of the actions taken in response to the instruction.
– You SHOULD NOT make assumptions based on information not presented in the screenshot when comparing it to the instructions.
– Your primary responsibility is to conduct a thorough assessment of the web task instruction against the outcome depicted in the screenshot and in the response, evaluating whether the actions taken align with the given instructions.
– NOTE that the instruction may involve more than one task, for example, locating the garage and summarizing the review. Failing to complete either task, such as not providing a summary, should be considered unsuccessful.
– NOTE that the screenshot is authentic, but the response provided by LLM is generated at the end of web browsing, and there may be discrepancies between the text and the screenshots.
– Note that if the content in the Result response is not mentioned on or different from the screenshot, mark it as not success.
– NOTE that the task may be impossible to complete, in which case the agent should indicate this in the response. CAREFULLY VERIFY THE SCREENSHOT TO DETERMINE IF THE TASK IS IMPOSSIBLE TO COMPLETE. Be aware that the agent may fail because of its incorrect actions, please do not mark it as impossible if the agent fails because of its incorrect actions.
You should explicit consider the following criterion:
- Whether the claims in the response can be verified by the screenshot. E.g. if the response claims the distance between two places, the screenshot should show the direction. YOU SHOULD EXPECT THAT THERE IS A HIGH CHANCE THAT THE AGENT WILL MAKE UP AN ANSWER NOT VERIFIED BY THE SCREENSHOT.
- Whether the agent completes EXACTLY what the task asks for. E.g. if the task asks to find a specific place, the agent should not find a similar place.
In your responses:
You should first provide thoughts EXPLICITLY VERIFY ALL THREE CRITERION and then provide a definitive verdict on whether the task has been successfully accomplished, either as 'SUCCESS' or 'NOT SUCCESS'.
A task is 'SUCCESS' only when all of the criteria are met. If any of the criteria are not met, the task should be considered 'NOT SUCCESS'.

## F.2  Agent Prompt

**WebVoayager**

Imagine you are a robot browsing the web, just like humans. Now you need to complete a task. In each iteration, you will receive an observation that includes the accessibility tree of the webpage and a screenshot of the current viewpoint. The accessbility tree contains information about the web elements and their properties. The screenshot will feature numerical labels placed in the TOP LEFT corner of web elements in the current viewpoint. Carefully analyze the webpage information to identify the numerical label corresponding to the web element that requires interaction, then follow the guidelines and choose one of the following actions:
1. Click a web element.
2. Delete existing content in a textbox and then type content.
3. Scroll up or down the whole window.
4. Go back, returning to the previous webpage.
5. Navigate to Bing's homepage.
6. Answer. This action should only be chosen when all questions in the task have been solved.
Correspondingly, action should STRICTLY follow the format specified by one of the following lines:
Click [numerical_label]
Type [numerical_label] [content]
Scroll [up/down]
GoBack

Bing
ANSWER [content]
Some examples are:
Click [8]
Type [22] [Boston]
Scroll [down]
Bing
ANSWER [06516]
Key guidelines you MUST follow:
* Action guidelines *
1. The predicted action should be based on elements as long as it's accessibility tree OR screenshot. Sometimes, accessibility tree or screenshot captures more elements than the other, but it's fine to use either one.
2. To input text for search bars, no need to click textbox first, directly type content. After typing, the system automatically hits 'ENTER' key.
3. When a complex task involves multiple questions or steps, select 'ANSWER' only at the very end, after addressing all of these questions or steps. Double check the formatting requirements in the task when ANSWER. Always think twice before using 'ANSWER' action!!!
4. When specifying the content for 'Type' and 'ANSWER' actions, be sure to wrap the content with '[]'.
5. Use 'GoBack' to return to the previous state, use it when you find the previous action incorrect.
6. When you see a pop-up page, you should immediately 'GoBack' to the previous page.
7. Use 'Bing' when you need to navigate to a different website or search for new information.
Your reply should strictly follow the format:
Thought: Your reasoning trace. A good practice is to follow this format:
- Observation summary: where are you at now? list all elements that are related to the task goal. e.g. if you're trying to filter something out, list all filters visible.
- Planning: what sequence of actions do you need take to achieve the task goal? give a high-level overview of the steps you need to take.
- Possible actions: to achieve that plan, what are potential actions you need to do immediately and what's their effect? List at least 3 actions and analyze each of them.
Action: Based on this reasoning, identify the single most optimal action. You should output it in the format specified above ("...STRICTLY follow the format...").
After you issue an action, the user will execute it and provide a new observation. Now solve the following task.
Task: {task_goal}
Current URL: {url}
Screenshot of current viewpoint: attached
Accessibility tree of current viewpoint: {accessibility_tree}

## F.3 Experiment Details

We use the following hyperparameters to obtain the WebVoyager results.

- num_iteration: 10
- actor_epochs: 1 # number of epochs to update the actor
- rollout_size: 512
- num_update_sample_per_iteration: 512
- lr: 1e-5
- optimizer: AdamW
- scheduler: WarmupCosineLR
- batch_size: 4
- grad_accum_steps: 2

- eval_horizon: 30 # note that train horizon is different for different methods, but evaluation horizon is kept the same

## F.4   Case Studies: Strengths 1

Task: Locate a recipe for an American apple pie on Allrecipes with a rating of at least 4 stars and more than 50 reviews. Note the maximum temperature mentioned in the Directions.

Fully trained agent explores:

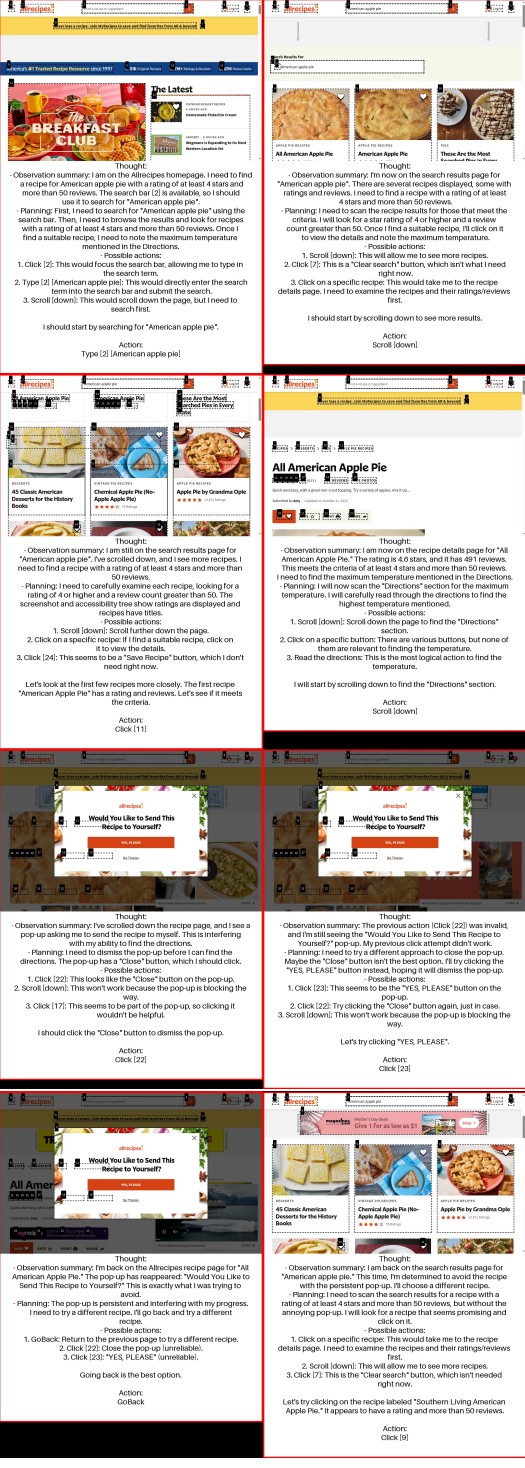

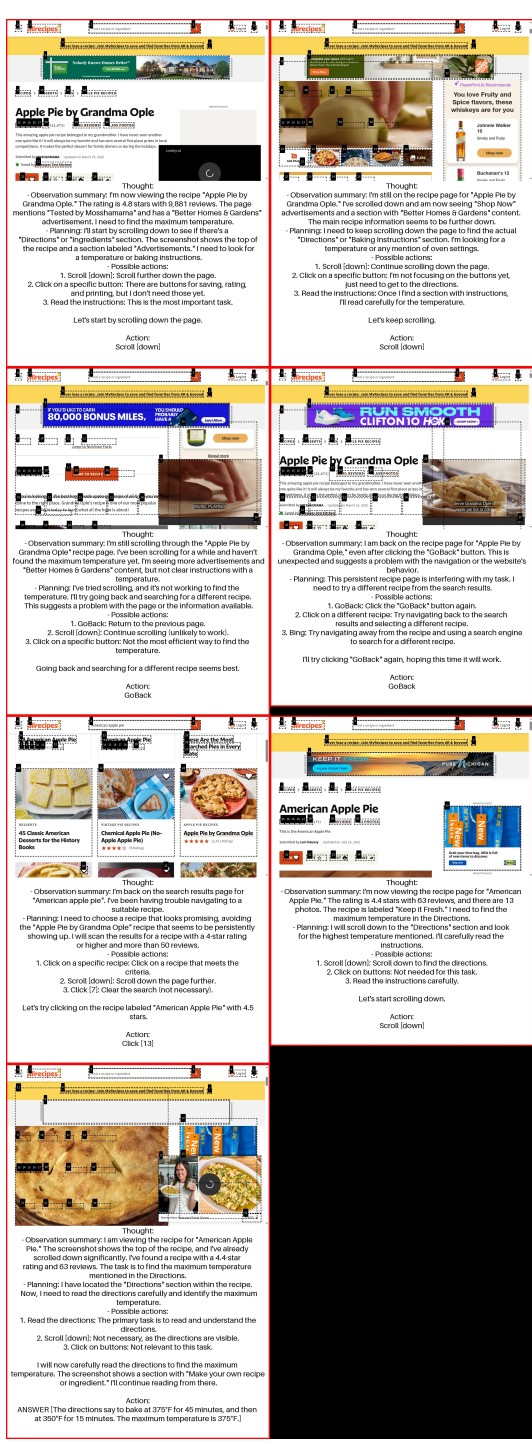

Thought:
- Observation summary: I'm now viewing the recipe "Apple Pie by Grandma Ople." The rating is 4.8 stars with 9,881 reviews. The page mentions "Tested by Mosshamama" and has a "Better Homes & Gardens" advertisement. I need to find the maximum temperature.
- Planning: I'll start by scrolling down to see if there's a "Directions" or "Ingredients" section. The screenshot shows the top of the recipe and a section labeled "Advertisements." I need to look for a temperature or baking instructions.
  - Possible actions:
    1. Scroll [down]: Scroll further down the page.
    2. Click on a specific button: There are buttons for saving, rating, and printing, but I don't need those yet.
    3. Read the instructions: This is the most important task.

Let's start by scrolling down the page.

Action:
Scroll [down]

Thought:
- Observation summary: I'm still on the recipe page for "Apple Pie by Grandma Ople." I've scrolled down and am now seeing "Shop Now" advertisements and a section with "Better Homes & Gardens" content. The main recipe information seems to be further down.
- Planning: I need to keep scrolling down the page to find the actual "Directions" or "Baking Instructions" section. I'm looking for a temperature or any mention of oven settings.
  - Possible actions:
    1. Scroll [down]: Continue scrolling down the page.
    2. Click on a specific button: I'm not focusing on the buttons yet, just need to get to the directions.
    3. Read the instructions: Once I find a section with instructions, I'll read carefully for the temperature.

Let's keep scrolling.

Action:
Scroll [down]

Thought:
- Observation summary: I'm still scrolling through the "Apple Pie by Grandma Ople" recipe page. I've been scrolling for a while and haven't found the maximum temperature yet. I'm seeing more advertisements and "Better Homes & Gardens" content, but not clear instructions with a temperature.
- Planning: I've tried scrolling, and it's not working to find the temperature. I'll try going back and searching for a different recipe. This suggests a problem with the page or the information available.
  - Possible actions:
    1. GoBack: Return to the previous page.
    2. Scroll [down]: Continue scrolling (unlikely to work).
    3. Click on a specific button: Not the most efficient way to find the temperature.

Going back and searching for a different recipe seems best.

Action:
GoBack

Thought:
- Observation summary: I am back on the recipe page for "Apple Pie by Grandma Ople," even after clicking the "GoBack" button. This is unexpected and suggests a problem with the navigation or the website's behavior.
- Planning: This persistent recipe page is interfering with my task. I need to try a different recipe from the search results.
  - Possible actions:
    1. GoBack: Click the "GoBack" button again.
    2. Click on a different recipe: Try navigating back to the search results and selecting a different recipe.
    3. Bing: Try navigating away from the recipe and using a search engine to search for a different recipe.

I'll try clicking "GoBack" again, hoping this time it will work.

Action:
GoBack

Thought:
- Observation summary: I'm back on the search results page for "American apple pie". I've been having trouble navigating to a suitable recipe.
- Planning: I need to choose a recipe that looks promising, avoiding the "Apple Pie by Grandma Ople" recipe that seems to be persistently showing up. I will scan the results for a recipe with a 4-star rating or higher and more than 50 reviews.
  - Possible actions:
    1. Click on a specific button: Click on a recipe that meets the criteria.
    2. Scroll [down]: Scroll down the page further.
    3. Click [7]: Clear the search (not necessary).

Let's try clicking on the recipe labeled "American Apple Pie" with 4.5 stars.

Action:
Click [13]

Thought:
- Observation summary: I'm back on the recipe page for "American Apple Pie." The rating is 4.4 stars with 63 reviews, and there are 13 photos. The recipe is labeled "Keep it Fresh." I need to find the maximum temperature in the Directions.
- Planning: I will scroll down to the "Directions" section and look for the highest temperature mentioned. I'll carefully read the instructions.
  - Possible actions:
    1. Scroll [down]: Scroll down to find the directions.
    2. Click on buttons: Not needed for this task.
    3. Read the instructions carefully.

Let's start scrolling down.

Action:
Scroll [down]

Thought:
- Observation summary: I am viewing the recipe for "American Apple Pie." The screenshot shows the top of the recipe, and I've already scrolled down significantly. I've found a recipe with a 4.4-star rating and 63 reviews. The task is to find the maximum temperature mentioned in the Directions.
- Planning: I have located the "Directions" section within the recipe. Now, I need to read the directions carefully and identify the maximum temperature.
  - Possible actions:
    1. Read the directions: The primary task is to read and understand the directions.
    2. Scroll [down]: Not necessary, as the directions are visible.
    3. Click on buttons: Not relevant to this task.

I will now carefully read the directions to find the maximum temperature. The screenshot shows a section with "Make your own recipe or ingredient." I'll continue reading from there.

Action:
ANSWER [The directions say to bake at 375°F for 45 minutes, and then at 350°F for 15 minutes. The maximum temperature is 375°F.]

Early-stage agent prefers exploitation:

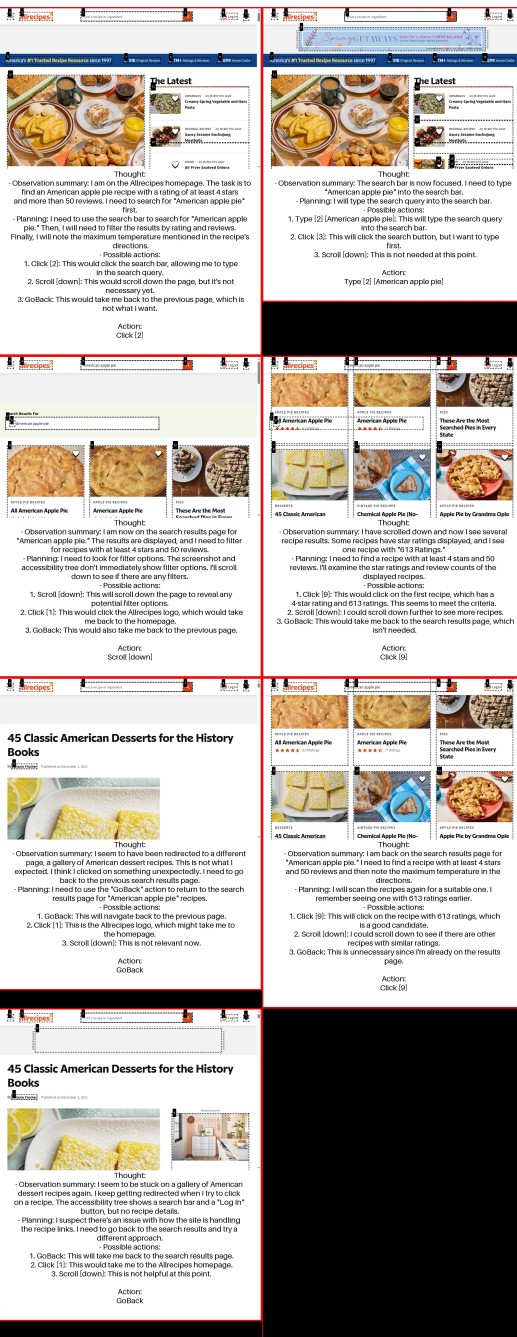

## F.5    Case Studies: Strengths 2

Task: Identify the latest top-trending open-source project in the category of 'Machine Learning' on GitHub, and check the number of stars it has received.

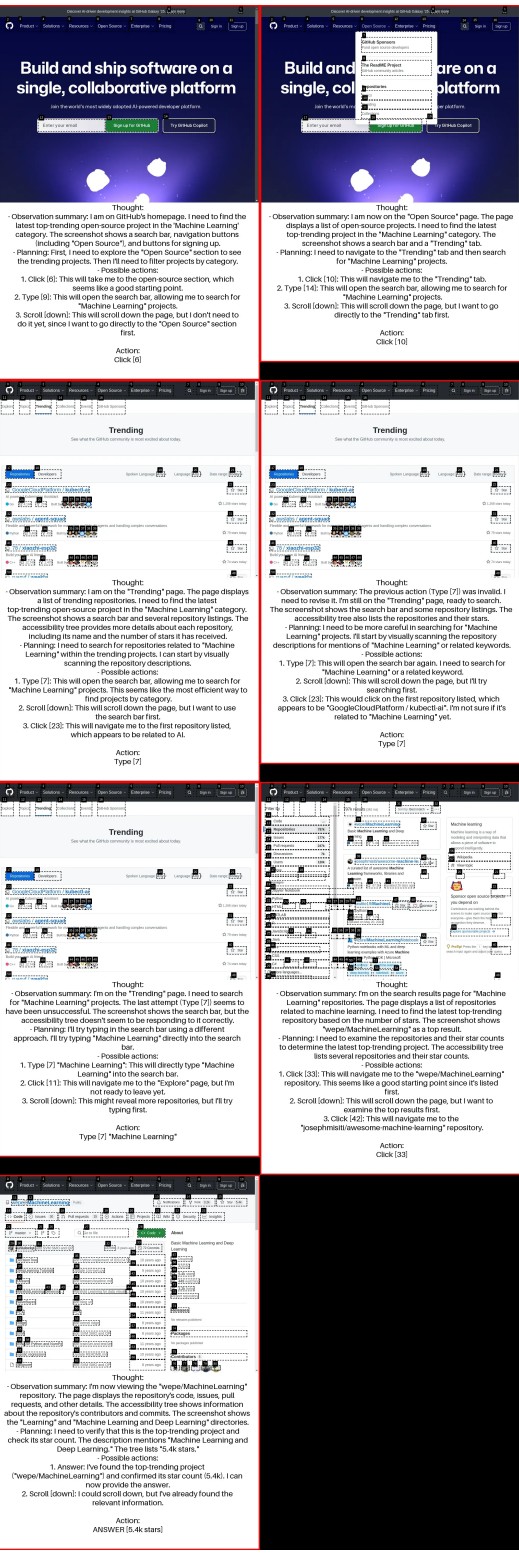

## F.6    Case Studies: Fail Modes 1

Task: On Apple's website, how many different types of keyboards are available when customizing your 14-inch MacBook Pro?

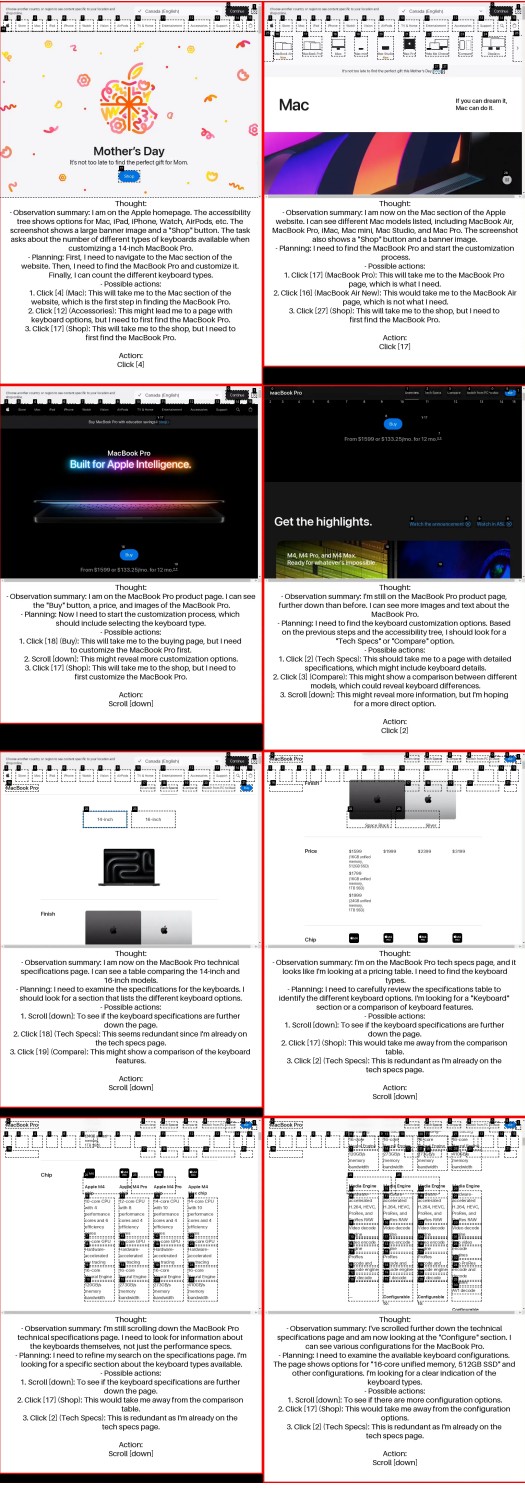

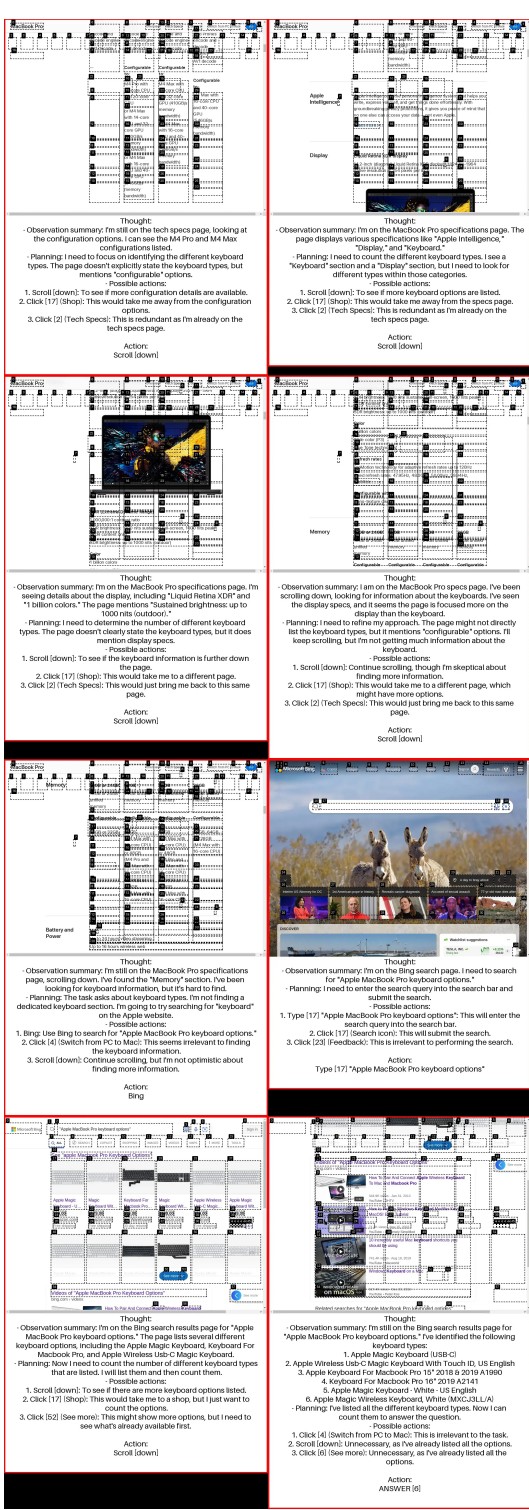

**Panel 1**

Thought:
- Observation summary: I'm still on the tech specs page, looking at the configuration options. I can see the M4 Pro and M4 Max configurations listed.
- Planning: I need to focus on identifying the different keyboard types. The page doesn't explicitly state the keyboard types, but mentions "configurable" options.
- Possible actions:
1. Scroll [down]: To see if more configuration details are available.
2. Click [17] [Shop]: This would take me away from the configuration options.
3. Click [2] [Tech Specs]: This is redundant as I'm already on the tech specs page.

Action:
Scroll [down]

**Panel 2**

Thought:
- Observation summary: I'm on the MacBook Pro specifications page. The page displays various specifications like "Apple Intelligence," "Display," and "Keyboard."
- Planning: I need to count the different keyboard types. I see a "Keyboard" section and a "Display" section, but I need to look for different types within those categories.
- Possible actions:
1. Scroll [down]: To see if more keyboard options are listed.
2. Click [17] [Shop]: This would take me away from the specs page.
3. Click [2] [Tech Specs]: This is redundant as I'm already on the tech specs page.

Action:
Scroll [down]

**Panel 3**

Thought:
- Observation summary: I'm on the MacBook Pro specifications page. I'm seeing details about the display, including "Liquid Retina XDR" and "1 billion colors." The page mentions "Sustained brightness: up to 1000 nits (outdoor)."
- Planning: I need to determine the number of different keyboard types. The page doesn't clearly state the keyboard types, but it does mention display specs.
- Possible actions:
1. Scroll [down]: To see if the keyboard information is further down the page.
2. Click [17] [Shop]: This would take me to a different page.
3. Click [2] [Tech Specs]: This would just bring me back to this same page.

Action:
Scroll [down]

**Panel 4**

Thought:
- Observation summary: I am on the MacBook Pro specs page. I've been scrolling down, looking for information about the keyboards. I've seen the display specs, and it seems the page is focused more on the display than the keyboard.
- Planning: I need to refine my approach. The page might not directly list the keyboard types, but it mentions "configurable" options. I'll keep scrolling, but I'm not getting much information about the keyboard.
- Possible actions:
1. Scroll [down]: Continue scrolling, though I'm skeptical about finding more information.
2. Click [17] [Shop]: This would take me to a different page, which might have more options.
3. Click [2] [Tech Specs]: This would just bring me back to this same page.

Action:
Scroll [down]

**Panel 5**

Thought:
- Observation summary: I'm on the MacBook Pro specifications page, scrolling down. I've found the "Memory" section. I've been looking for keyboard information, but it's hard to find.
- Planning: The task asks about keyboard types. I'm not finding a dedicated keyboard section. I'm going to try searching for "keyboard" on the Apple website.
- Possible actions:
1. Bing: Use Bing to search for "Apple MacBook Pro keyboard options."
2. Click [4] [Switch from PC to Mac]: This seems irrelevant to finding the keyboard information.
3. Scroll [down]: Continue scrolling, but I'm not optimistic about finding more information.

Action:
Bing

**Panel 6**

Thought:
- Observation summary: I'm on the Bing search page. I need to search for "Apple MacBook Pro keyboard options."
- Planning: I need to enter the search query into the search bar and submit the search.
- Possible actions:
1. Type [17] "Apple MacBook Pro keyboard options": This will enter the search query into the search bar.
2. Click [17] [Search icon]: This will submit the search.
3. Click [23] [Feedback]: This is irrelevant to performing the search.

Action:
Type [17] "Apple MacBook Pro keyboard options"

**Panel 7**

Thought:
- Observation summary: I'm on the Bing search results page for "Apple MacBook Pro keyboard options." The page lists several keyboard options, including the Apple Magic Keyboard, Keyboard For Macbook Pro, and Apple Wireless Usb-C Magic Keyboard.
- Planning: Now I need to count the number of different keyboard types that are listed. I will list them and then count them.
- Possible actions:
1. Scroll [down]: To see if there are more keyboard options listed.
2. Click [17] [Shop]: This would take me to a shop, but I just want to count the options.
3. Click [52] [See more]: This might show more options, but I need to see what's already available first.

Action:
Scroll [down]

**Panel 8**

Thought:
- Observation summary: I'm still on the Bing search results page for "Apple MacBook Pro keyboard options." I've identified the following keyboard types:
1. Apple Magic Keyboard (USB-C)
2. Apple Wireless Usb-C Magic Keyboard With Touch ID, US English
3. Apple Keyboard For Macbook Pro 15" 2018 & 2019 A1990
4. Keyboard For Macbook Pro 16" 2019 A2141
5. Apple Magic Keyboard - White - US English
6. Apple Magic Wireless Keyboard, White (MXCJ3LL/A)
- Planning: I've listed all the different keyboard types. Now I can count them to answer the question.
- Possible actions:
1. Click [4] [Switch from PC to Mac]: This is irrelevant to the task.
2. Scroll [down]: Unnecessary, as I've already listed all the options.
3. Click [6] [See more]: Unnecessary, as I've already listed all the options.

Action:
ANSWER [6]

Task: Which university maintains and manages ArXiv. Accessing the university's website from ArXiv, how many underegraduate students are currently at the university.

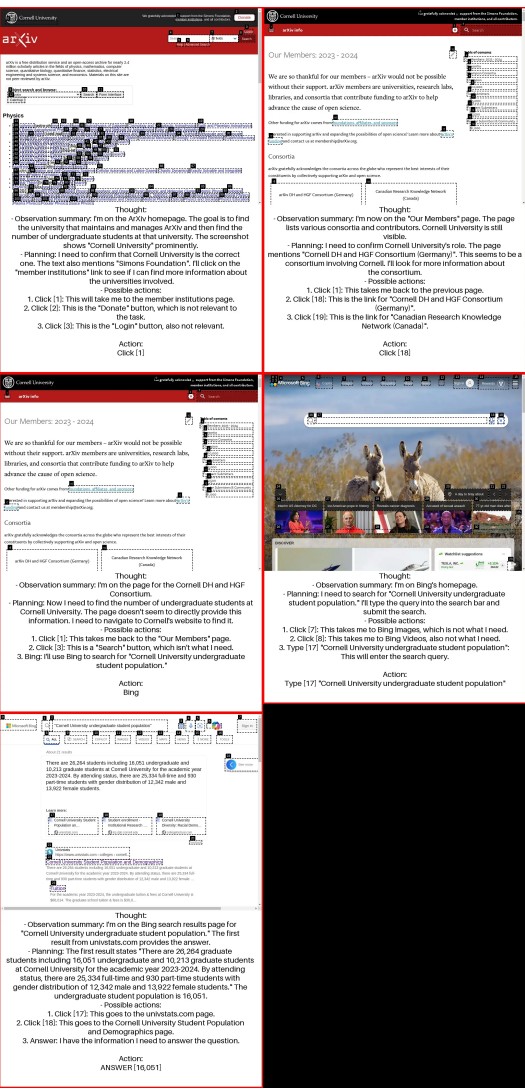

### F.7 Case Studies: Fail Modes 2

Task: Identify a new open-source project on GitHub related to 'AI agriculture' that created in 2022, and note its main programming language and description.

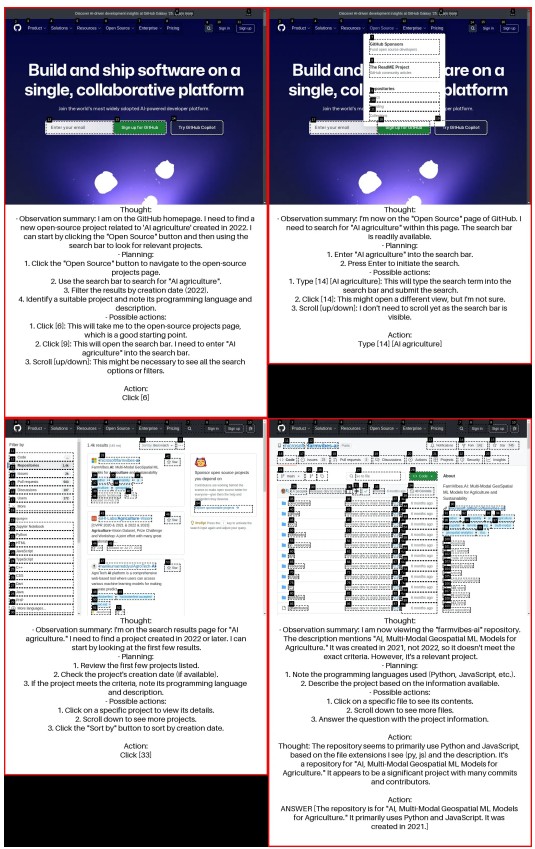

## G Full WebArena Experiment Details

We use the following hyperparameters to obtain the full WebArena results.

- num_iteration: 10
- actor_epochs: 1 # number of epochs to update the actor
- rollout_size: 512
- num_update_sample_per_iteration: 512
- lr: 1e-6
- optimizer: AdamW
- scheduler: WarmupCosineLR
- batch_size: 4
- grad_accum_steps: 2
- eval_horizon: 30

