# OpenReview forum: "Thinking vs. Doing: Improving Agent Reasoning by  Scaling Test-Time Interaction"
_NeurIPS.cc/2025/Conference — NeurIPS 2025 poster_

### Official Review · Reviewer_13jF · 2025-06-10

**Clarity:** 3
**Significance:** 4
**Originality:** 4
**Rating:** 5
**Confidence:** 4

**Summary:**

The authors investigate different ways to scale up inference compute for web agents. Interestingly, per-step inference compute  does not improve performance as much as longer interactions with the environment. Longer episodes allow the agent to collect more information to achieve its tasks. While it is possible to prompt an agent to perform more interactions, it is not obvious how to teach one using RL. Surprisingly, the authors show that a simple curriculum allow the agent to learn how to interact longer with a task to succeed.

**Questions:**

* Do you plan to train your agent with a more sophisticated method than online behavior cloning? Long horizon tasks such as web browsing could benefit greatly from a method that performs credit assignment such as PPO.
* The drop in performance in Figure 4 as training steps increase is bizarre. Can you explain why the performance drops on all domains for interaction 5 and 10?

**Ethical Concerns:**

["NO or VERY MINOR ethics concerns only"]

**Final Justification:**

Test time compute has been very popular in the recent months and this paper provide a fresh perspective on how it should be used for browser based environments. The method is simple and carefully evaluated, therefore I do recommend to accept the paper.

**Limitations:**

yes

**Quality:**

3

**Strengths And Weaknesses:**

# Strengths
## Simplicity
* Despite the simplicity of curriculum learn and online filtered behavior cloning, the resulting agent performs favorably compare to the baselines.

## Empirical performance
* Figure 2 validates both the need to have longer episodes and the need for training, since performance improvement is task dependent.
* Table 3 shows clear improvement of TTI over the baselines. The improvement is smaller for WebArena, but still positive.

## The ablations are interesting
* In section 4.2, it is shown that per step test time compute (best of n and budget forcing) do not work as well as interaction scaling.

# Weaknesses

## Presentation
* The legend in Figure 6 is missing the circles.
* What environment is Table 3? WebVoyager?

## Compute cost
* Compute cost is brought up constantly in the paper, would it be possible to include them?

# Minor
Typo: Comaprisons.

---

> ### Author Rebuttal · Authors · 2025-07-30
>
> We sincerely thank the reviewer for the positive and thoughtful feedback. We're glad that you found our approach simple yet effective, and we appreciate your recognition of the paper’s originality, significance, and empirical contributions. To address your question regarding computational costs, **we add additional analysis on the token count, number of forward passes, and training time of TTI**. Below, we provide a detailed response to each comment.
>
>
> ---
>
> ## Weakness 2: Compute Cost
>
> > Compute cost is brought up constantly in the paper, would it be possible to include them?
>
> Thank you for this suggestion. While we reported token-level compute costs in Figure 3 (top), we agree that a more detailed breakdown for large-scale training would be helpful. We update Appendix F.3 to include the following table for the WebVoyager experiments (Table 3). All settings use Gemma 3 12B. Statistics computed on the training set. Training time includes both rollout collection and model update performed on 4 H100s for 10 iterations.
>
>
> | Method         | Avg Total Tokens per Rollout | Avg Model Calls (# Steps) per Rollout | Training Wallclock Time (hrs) |
> |----------------|--------------------------|------------------------------|-------------------------------|
> | Baseline (Prompting Only)    |        810         |         10.8        |              -         |
> | Fixed h = 10   |     700       |            9.5      |       10            |
> | Fixed h = 30   |      760            |         14.1           |         25              |
> | TTI    |      640          |       10.5       |       20               |
>
>
> In summary, TTI improves success rate while requiring fewer training compute than fixed h=30 baselines, making it both more effective and efficient in our current setup.
>
> ---
>
> ## Question 2: Fixed-Horizon Baseline Performance
>
> > The drop in performance in Figure 4 as training steps increase is bizarre. Can you explain why the performance drops on all domains for interaction 5 and 10?
>
> We hypothesize that the drop in performance for short-horizon agents is due to:
> - Early overfitting: With smaller horizons, the agent initially learns to exploit short paths but quickly overfits to local patterns that do not generalize well on the held-out set.
> - Limited rollout diversity: These settings fail to expose the agent to exploratory trajectories or recovery strategies, leading to a plateau or even regression as training progresses.
> - Reduced data coverage: Since each rollout is shorter, fewer states are visited overall, reducing the diversity of successful examples stored in the replay buffer.
>
> We will add discussion of these dynamics in Section 5.1. The drops in fixed-horizon training actually motivate our curriculum approach. TTI (Figure 6) shows more stable learning because its curriculum helps mitigate this collapse by exposing the agent to longer interactions progressively, improving both generalization and robustness.
>
> ---
>
> ## Question 1: Future Work
>
> > Do you plan to train your agent with a more sophisticated method than online behavior cloning? Long horizon tasks such as web browsing could benefit greatly from a method that performs credit assignment such as PPO.
>
> Yes, we agree that more advanced RL algorithms such as PPO, GRPO, or Q-learning-based methods could offer improved credit assignment over online filtered behavior cloning, particularly for long-horizon tasks with sparse rewards. We chose to begin with a simple and robust baseline to isolate the effects of curriculum over interaction horizon, but we absolutely see value in applying policy-gradient-based methods or even value-based methods in future work.
>
> As noted in Section 7, combining TTI with more advanced RL methods is a promising direction. In particular, using PPO could help stabilize training with longer horizons and enable more nuanced learning of exploration strategies. We plan to explore this in the future.
>
>
> ---
>
> ## Weakness 1: Presentation
>
> Figure 6 legend: Thank you for pointing this out. We have fixed the missing legend markers in the updated version.
>
> Table 3 environment: Table 3 reports results on WebVoyager (we now explicitly clarify this in the caption and surrounding text).
>
> Typo (“Comaprisons”): Fixed—thank you for catching this.

---

### Official Review · Reviewer_GJf2 · 2025-06-29

**Clarity:** 2
**Significance:** 1
**Originality:** 1
**Rating:** 2
**Confidence:** 4

**Summary:**

The paper tackles the often-overlooked problem of scaling an agent’s interaction count rather than merely its per-step reasoning. A simple “re-check” prompt shows that allowing a Gemma-3-12B web agent to take extra actions lifts WebArena success rates. Building on this observation, the authors introduce Test-Time Interaction (TTI)—an online RL recipe that (i) filters trajectories to keep only successful ones and (ii) trains under a curriculum that gradually expands the maximum number of steps per episode. On the WebVoyager and WebArena benchmarks, a TTI-fine-tuned Gemma-3-12B sets a state-of-the-art among fully open-source agents.

**Questions:**

1. Can TTI be used in general tasks and environments where each action is risky or costly and, if so, what changes would it need?
2. How can we keep extra actions from becoming too expensive or dangerous while still reaping the benefits of longer interaction?

**Ethical Concerns:**

["NO or VERY MINOR ethics concerns only"]

**Final Justification:**

I appreciate the authors' efforts to address my concerns. However, I am maintaining my original score for the following reasons:

- I am still not convinced that test-time interaction scaling is a general and promising direction.
- I find that the proposed method is neither significant nor original, especially given the authors’ rebuttal stating that the curriculum heuristic they used is a well-established practice.
- While the issue of the single-run experiment has been resolved, I believe this constitutes only a minor improvement compared to the aforementioned unresolved concerns.

**Limitations:**

yes

**Paper Formatting Concerns:**

I haven’t noticed any major formatting issues.

**Quality:**

2

**Strengths And Weaknesses:**

**Strengths**

1. The paper presents a new scaling axis—allowing more interaction steps rather than just deeper per-step reasoning.
2. It spells out the credit-assignment and gradient-signal problems that arise with long horizons and mitigates them through a horizon curriculum combined with reward-filtered updates.
3. The proposed method delivers empirical gains on web tasks.

**Weaknesses**

1. Extending the interaction horizon is a double-edged sword: every extra step incurs latency, compute, and energy overhead, but—more critically—the agent is given many more chances to do something harmful or irreversible. Even in seemingly “safe” web tasks, longer roll-outs increase the odds of clicking fraudulent ads, submitting a form twice, buying unwanted items, or spamming an email thread. In real-world domains (finance, healthcare, robotics) a single stray action can be catastrophic or legally non-compliant. Yet the paper leaves these risks unaddressed, creating a substantial gap for practical deployment.
2. The horizon-scheduling curriculum that tackles the training difficulty of longer interactions is still largely heuristic.
3. Main results come from a single run, so robustness to random seeds remains uncertain at full scale.

---

> ### Author Rebuttal · Authors · 2025-07-30
>
> We thank the reviewer for their thoughtful feedback and for recognizing the empirical gains of our method. To address your concern about the number of experimental trials, we now **include additional runs**, demonstrating that our results are statistically consistent. We believe that these results should help address concerns regarding the efficacy of our approach.
>
> ## At the outset, we would like to first address your concern that TTI is impractical because "extending the interaction horizon is a double-edged sword."
>
> At a high level, a similar argument can be made about any expressive agent/policy architecture: using a larger model, more interaction steps, or longer reasoning chains, could in principle represent policies that increase the probability of risky behavior or incur higher latency and compute costs. However, recent advances in reasoning and agentic tasks have demonstrated that with the right training objectives and workflows, we can guide such expressive policies toward useful solutions rather than pathological ones, leading to improved generalization and performance scaling.
>
> We developed TTI in the same spirit. **The RL-based training used in TTI explicitly discourages risky or degenerate behaviors** like submitting forms incorrectly or clicking fraudulent links, because these behaviors minimize reward  (see our RL objective in Eq. 1). Thus, TTI allows the model to benefit from extended interaction only when it leads to better outcomes. Case studies in Section 6.2 also confirm that TTI agents learn strategic behaviors like backtracking and verification, not random exploration. In fact, pretty much like how in closed-domain reasoning, using reasoning improves model safety (e.g., see deliberative alignment from OpenAI), we believe that deliberative behaviors that use more interaction steps should only help an agent detect risk early on through exploration and planning, while still enabling it to solve the task eventually.
>
> Like other test-time scaling techniques (e.g., long-chain CoT prompting in LLMs), **TTI does introduce some overhead, but in exchange, it yields significantly more robust and capable behavior,** especially on complex tasks where shorter horizons fall short. Indeed, just as long CoT traces have become widely adopted in LLM reasoning despite their compute cost, we view TTI as an analogous strategy for agentic settings, where interaction reveals new information critical for success. **We therefore believe it would be unfair to reject our work based solely on presumed risks or overhead, particularly in light of the benefits.**
>
> Below, we address other comments in detail.
>
> ---
>
> ## Weakness 3: Multiple Runs
>
> > Main results come from a single run, so robustness to random seeds remains uncertain at full scale.
>
> Thank you for raising this point. First, we want to clarify that **results in Section 4 and 5.2 are already averaged over three runs**, as we noted in the caption of Figure 1 "results averaged over 3 runs." Since we only include the average numbers in the paper, we also provide the error bars below, which we will add to the appendix.
> |Method|Success Rate on WebArena Subset(%)|
> |-|-|
> |Baseline|23.81±1.6|
> |Fixed h=5|24.59±1.5|
> |Fixed h=10|25.19±1.2|
> |Fixed h=20|29.28±1.2|
> |TTI Additive Schedule|29.5±1.4|
> |TTI Multiplicative Schedule|32.35±1.3|
>
> For large-scale RL training on WebVoyager and WebArena (Table 3 & 4), we note that the training tasks are over 100K, making rollout collection super expensive. Due to resource constraints, we thus prioritized domain and benchmark coverage over running multiple seeds. We note that recent work on web agents that are published in conferences, such as Learn-By-Interact (ICLR 2025), AgentOccam (ICLR 2025), WebRL (ICLR 2025), and Agent Workflow Memory (ICML 2025), all report single-run results on large-scale benchmarks due to the cost of training and evaluation. We agree that adding multiple seeds is valuable and will include this in the final version, though we expect our results and conclusions to be solid.
>
> Finally, several other reviewers (kwcd, 13jF) independently noted the consistency of performance across 20+ domains and 2 benchmarks. Besides, our results can be  reproduced with the released codebase and training details, and we plan to open-source checkpoints for verification. These all improve the reproducibility and robustness of our work.
>
> ---
> ## Weakness 2: Curriculum Heuristic
>
> > The horizon-scheduling curriculum that tackles the training difficulty of longer interactions is still largely heuristic.
>
> While our curriculum does not emerge from a theoretical derivation, **it is grounded in well-established practices from both RL and LLM training rather than merely being ad-hoc heuristics**. Specifically,
>
> **Horizon scaling** has been shown effective in prior work such as CodeT [1] and Reflexion [2], though none of these works have proposed a formal training mechanism. Our horizon-based schedule mirrors established RL curricula [3, 4] that gradually increase task difficulty to improve stability under sparse rewards.
>
> As for the specific **curriculum schedule**, our use of a multiplicative curriculum (Eq. 3) is not purely heuristic because: (1) multiplicative schedule is very common in existing reasoning works; (2) we study several design choices such as additive schedule as ablations. Both of them outperform baseline agents and test-time-only strategies like budget forcing or best-of-n prompting (Table 2 and Figure 3). However, the multiplicative schedule is empirically more effective, so we adopt it for later experiments; (3) on WebVoyager,  TTI outperforms fixed-horizon baselines (Figure 4, 6). Such strong downstream results show the effectiveness of our curriculum design.
>
> [1] CodeT: Code Generation with Generated Tests. ICLR 2023.
>
> [2] Reflexion: Language Agents with Verbal Reinforcement Learning. NeurIPS 2023.
>
> [3] WebRL: Training llm web agents via self-evolving online curriculum reinforcement learning. ICLR 2025.
>
> [4] AutoWebGLM: A large language model-based web navigating agent.  KDD 2024.
>
> ---
>
> ## Question: Applicability to Risky Settings
>
> >  Can TTI be used in general tasks and environments where each action is risky or costly and, if so, what changes would it need?
>
> > How can we keep extra actions from becoming too expensive or dangerous while still reaping the benefits of longer interaction?
>
> Yes, TTI can be adapted to general environments where actions are risky or costly as future work. For example, we can introduce cost-aware curricula that assign higher weights to risky actions and optimize performance subject to total cost constraints, similar to safe RL frameworks [1].  We can also enforce risk-bounded exploration by limiting high-impact actions during early training, and gradually relaxing constraints.  For example, in robotics, TTI could first allow only low-cost actions such as camera adjustments, then gradually introduce manipulation steps, with irreversible actions gated behind explicit confirmation.
>
> As for balancing longer interaction and the quality of each action, our findings in the paper already suggest a natural division of labor between internal reasoning and external action: internal "thinking" for planning and verification, and external interaction for exploration and information gathering. Thus, we can develop hybrid test-time scaling, where agents learn to dynamically allocate compute based on environmental cost and task uncertainty to achieve robust performance gains. We will add the above discussion to Section 7.
>
> These extensions are important directions for applying TTI to more real-world settings, but are beyond the current scope of our work. We think it would be unfair to reject our paper based on the presumed risks in diverse environments.
>
> [1] A Review of Safe Reinforcement Learning: Methods, Theory and Applications, Gu et al., 2024.

---

> > ### Author Response · Authors · 2025-08-03
> > **Kind Reminder to Discuss**
> >
> > Dear Reviewer,
> >
> > We apologize for bothering you, but since the discussion period is very short, we wanted to check in with you to see if our rebuttal addresses all outstanding concerns and have a chance to address any new ones. Thank you!

---

> > ### Comment · Reviewer_GJf2 · 2025-08-05
> >
> > I appreciate the authors’ efforts in addressing my concerns.
> >
> > While the authors present TTI as one of “expressive agent/policy architectures,” I believe it is important to distinguish between *increasing the number of interaction steps* and *using a larger model* or *longer reasoning chains*. The latter can be performed independently of the environment, whereas the former inherently increases the chance of performing harmful or irreversible behaviors. This distinction is significant, and I believe it underlies why many prior works in the RL domain report performance relative to environment steps or under constrained interaction budgets.
> >
> > Regarding the authors’ claim that the RL-based training used in TTI “explicitly discourages risky or degenerate behaviors,” I would like to emphasize that in general agentic domains, enabling unrestricted action exploration through RL can lead to unintended consequences unless the system is carefully designed. Even if such behaviors can be discouraged through the use of specific objective functions, the agent must still engage in the risky behavior at least once to experience the associated negative reward. While I appreciate the mention of safe RL methods and risk-bounded exploration, I believe these aspects deserve a more central and thorough discussion within the scope of this work.
> >
> > I thank the authors for addressing weaknesses 2 and 3, which I consider partially resolved. However, my primary concern has not been fully resolved. Therefore, I believe my score should remain unchanged.

---

> > > ### Author Response · Authors · 2025-08-05
> > > **Followup**
> > >
> > > Thank you for engaging with our rebuttal! **We would first like to ask—what would precisely address your concerns regarding weakness 2 and 3?** We already added results with multiple runs despite the cost of running such an evaluation, and explained the rationale behind a curriculum including a discussion of other prior work that has built curriculum strategies intuitively. We are therefore unable to understand what clarifications/results would fully address these points. Would you mind elaborating on this aspect?
> > >
> > > In regards to the followup concerns, please see response below.
> > >
> > > ---
> > > ### Regarding the number of interaction steps and expressive agent/policy architectures
> > >
> > > First, we clarify an important distinction between (i) performance as a function of total environment steps seen during training (i.e., sample efficiency), and (ii) the number of interaction steps within a single trajectory. In traditional RL, we care about sample efficiency, i.e., how well a policy performs given a fixed interaction budget throughout training. This is distinct from per-trajectory interaction length, and in fact, most sample-efficient RL methods aim to reduce the number of trajectories needed, not necessarily their length. Standard RL is framed in an infinite-horizon setting, where trajectory lengths are $\infty$ (far longer than what TTI uses), yet the goal remains to optimize performance under a limited sample budget. In empirical work, a finite (but fixed) trajectory length is used simply for tractability.
> > >
> > > We clarify that TTI does not aim to increase the number of trajectories used for learning, but it simply expands the horizon length of a single trajectory during training, which is different from the setting you mention. That is, we want to still learn policies that could act for longer in an episode, while being sample efficient. We indeed see this being the case in our experiments in Table 3 and Figure 5.  **We are unaware of any RL work that claims to improve sample-efficiency by shortening the length of the episode.**
> > >
> > > **As for the connection between increased interaction steps and more expressive policies, this ties directly into the theoretical and conceptual benefits of learning richer POMDP policies.** Longer per-trajectory horizons enable agents to express more nuanced behavior, such as process-of-elimination reasoning, retrying, or backtracking. These types of behaviors can only emerge in policies that condition on extended histories. Indeed, prior work [1] explicitly argues that learning history-conditioned policies improves generalization to unseen tasks. Our approach aligns with this view: by enabling longer interaction, we allow policies to acquire and execute more algorithmic behaviors, which in turn enhances generalization.
> > >
> > > ---
> > >
> > > ### Regarding safety and risky behaviors
> > >
> > > Thank you for the comment! We agree that safe RL methods warrant a more central discussion, and we'll add this thoroughly in the final version (we promised this in our initial response). We'll also add a disclaimer for practitioners noting that, unless carefully handled, scaling interactions with an untrained or underperforming model can indeed pose safety risks.
> > >
> > > That said, we believe the concern that TTI experiences “negative reward through uncontrolled exploration in RL” is misplaced. First, it is important to note that **_any_ RL method must encounter negative reward at some point during training to learn what not to do, this is not unique to TTI**. This is one of the reasons why agents for computer tasks are typically trained in sandboxed environments (e.g., WebArena, Terminal Bench) or why robotic policies are commonly trained in simulation before deployment on real systems.
> > >
> > > Second, our proposed curriculum approach is designed to address the issue of uncontrolled or degenerate exploration. Rather than directly training an agent with a large per-trajectory interaction budget, which may lead to unsafe or noisy behavior, our method first perfects short-horizon behaviors and then gradually composes them as the interaction budget increases. **This leads to more structured and meaningful exploration in a “high-level” space of behaviors**, as opposed to the random action sequences that can arise when training directly with long horizons. Moreover, when combined with a reward function that emphasizes safety, **TTI curriculum should be at least as safe as the standard approach of training with shorter interaction budgets, and much safer than methods that use a fixed long horizon from the start.**
> > >
> > > Finally, we would also like to note that even long CoT reasoning can in principle produce harder to monitor or unsafe CoTs despite no interaction with the environment.
> > >
> > > ---
> > >
> > > **Please let us know if this response addresses your concerns.** If not, what specific experiments/discussions you are looking for  to reconsider your evaluation. Thank you very much.
> > >
> > >
> > > [1] Why Generalization in RL is Difficult. Ghosh et al. NeurIPS 2021.

---

> > > > ### Comment · Area_Chair_mb2z · 2025-08-09
> > > >
> > > > Dear Reviewer GJf2,
> > > >
> > > > Thank you for the active discussions with the authors. We understand your concerns about the potential risky behaviors resulting from the proposed technique. Separately, could you please also assess whether the authors' extensive rebuttal have addressed your concerns regarding the significance (1 = poor) and originality (1 = poor) of the proposed method? Given less than 24 hours remain in the discussion period, the authors and AC would be grateful if you could jump in asap. Thank you!

---

### Official Review · Reviewer_kwcd · 2025-07-03

**Clarity:** 3
**Significance:** 4
**Originality:** 4
**Rating:** 5
**Confidence:** 4

**Summary:**

The paper introduces Test-Time Interaction (TTI), a curriculum-based online reinforcement-learning method designed to scale the interaction of LLMs. The authors apply TTI to fine-tune Gemma 3-12B for web navigation tasks on both WebArena and WebVoyager. By progressively lengthening the agent’s interaction horizon during training, the resulting TTI model ,on average, surpasses other open-weight baselines on the two benchmarks. The paper also supplies ablations on the components of TTI and exposes typical success modes and failure patterns.

**Questions:**

1. In the GitHub, ESPN, and Coursera domains in Table 3 the base Gemma 3-12B outperforms its TTI-trained counterpart. Do these domains share characteristics that explain the drop in performance when applying TTI?
2. The variant trained with a fixed horizon of h = 10 outperforms both the curriculum schedule and the h = 30 variant on several domains in Table 3. Why is this intermediate horizon especially effective for the Amazon, ArXiv, GitHub, and Coursera domains?

**Ethical Concerns:**

["NO or VERY MINOR ethics concerns only"]

**Limitations:**

The authors have adequately addressed the limitations.

**Paper Formatting Concerns:**

I didn't notice any major formatting issues.

**Quality:**

3

**Strengths And Weaknesses:**

### Strengths
- Sound experimental design.
- Case studies effectively illustrate both capabilities and typical failure modes.
- The paper shows empiirically that careful horizon curricula matter at least as much as model size in this setting.

### Weaknesses
- Adding per-domain training curves (success rate vs. training steps) would be helpful.

---

> ### Author Rebuttal · Authors · 2025-07-30
>
> We sincerely thank the reviewer for the thoughtful and encouraging feedback. We appreciate your recognition of the originality and significance of our work. To address your questions regarding  more visualizations, **we will add new per-domain training curves to the paper**. Below, we provide responses to each comment.
>
> ---
>
> ## Weakness 1
>
> > Adding per-domain training curves (success rate vs. training steps) would be helpful.
>
> This is valuable feedback that would strengthen our analysis significantly! Due to space constraints, we prioritized aggregate results and included training curves for representative domains (e.g., Amazon, HuggingFace) in Fig. 5 (right). However, we have now added **per-domain success rate trends across training iterations for all WebVoyager domains** in Appendix F.9 of the updated version and would be happy to highlight these further in the main paper.
>
> Since figures cannot be shown in the response (and neither can links), we summarize key findings below. The domains exhibit distinct learning dynamics, which we broadly group into three categories:
> - Fast-learning domains (e.g., BBC, Apple): Plateau after ~5–6 iterations.
> - Moderate-learning domains (e.g., Allrecipes, HuggingFace): Improve steadily over 8–10 iterations.
> - Prone-to-overfitting domains (e.g., GitHub, Amazon): Performance may decline in later stages, as the agent learns to explore more despite the test tasks being simple and not requiring exploration (see our response to Q1 below for further discussion).
>
> These differences appear to stem from domain complexity and the base model’s familiarity with each domain. We believe these insights are an important step toward understanding when and where TTI is most beneficial, and they point to promising future directions like tailoring curriculum schedules to domain difficulty or agent priors.
>
> ---
>
> ## Question 1
>
> > In the GitHub, ESPN, and Coursera domains in Table 3 the base Gemma 3-12B outperforms its TTI-trained counterpart. Do these domains share characteristics that explain the drop in performance when applying TTI?
>
> This is an excellent observation. In our analysis (Section 6.2 and Appendix F.6), we found that these domains are likely to be better represented in the base model's pretraining data, so the zero-shot agent already performs pretty good. In these cases, TTI’s added exploration can occasionally lead to unnecessary detours, analogous to how reasoning models can overthink on simple tasks. For example, the TTI agent may query external search tools when a direct filter suffices (GitHub), or compare multiple courses when one stands out clearly (Coursera). These are **recoverable inefficiencies of our approach**, not failures of the method.
>
> We view the performance in GitHub, ESPN, and Coursera positively and as motivation for developing a more adaptive system in the future. Akin to how reasoning models benefit from learning when to invoke long CoTs and when to produce a response  directly (referred to as “dual mode thinking”), we can improve TTI by dynamically allocating interaction vs. per-step compute based on task instruction features. We view this as an important next step of our work, that can build upon the training protocols we introduce or distill our agent.
>
> ---
>
> ## Question 2
>
> > The variant trained with a fixed horizon of h = 10 outperforms both the curriculum schedule and the h = 30 variant on several domains in Table 3. Why is this intermediate horizon especially effective for the Amazon, ArXiv, GitHub, and Coursera domains?
>
> The fixed horizon of h = 10 performs well in domains like Amazon, ArXiv, GitHub, and Coursera because many tasks in these settings have short, structured workflows, e.g., "find a paper by title," "open a GitHub repo," or "enroll in a top-rated course." In such cases, a longer horizon may introduce unnecessary exploration that distracts from the optimal path, similar to overthinking in CoT models.
> However, this is not a limitation of TTI, but **a reflection of its general-purpose design**. While fixed h = 10 happens to be well-matched to simple and straightforward tasks, TTI is trained to adaptively scale horizons across diverse domains. As shown in Figure 5, TTI matches or exceeds h = 10 on most domains and excels in information-dense or partially observable settings like Allrecipes or Cambridge, where longer interaction is essential.
>
> We believe future extensions of TTI could incorporate domain-aware horizon control to better align with task complexity. This is a future direction we plan to explore.

---

> > ### Comment · Reviewer_kwcd · 2025-08-08
> >
> > Thank you for your detailed response and the helpful clarifications. I appreciate the effort you put into addressing the points raised in the review. Based on your replies, I am satisfied with the explanations provided and I intend to keep my current evaluation.

---

### Official Review · Reviewer_nVXA · 2025-07-03

**Clarity:** 3
**Significance:** 2
**Originality:** 2
**Rating:** 4
**Confidence:** 3

**Summary:**

The paper introduces test-time interaction scaling as a complementary axis to the usual “think-longer” compute scaling for agentic tasks. By allowing an agent to act for more steps before termination, the authors show that the agent can gain new external information by simply continuing to interact with the environment once again.  They further propose TTI—a curriculum-based online RL method that gradually lengthens the rollout horizon with a multiplicative schedule. Empirically, a Gemma-3 12 B model trained with TTI attains 66.5% success on WebVoyager and 26.1% on WebArena. Case studies illuminate how the agent learns to trade off exploration vs. exploitation and reveal residual failure modes.

**Questions:**

1. While the paper shows interaction horizon is increased with RL training under TTI curriculum, it is not clear how the agent learns to think longer during training. For instance, in Fig 4, after 400 steps training, the agent (h=20) stop increasing overall interaction horizon but still improves the performance. Could you clarify this and the saying in L248, and provide the thinking lengths?
2. In Fig 6, the "check-again" strategy seems to be less effective to increase the trajecory lengths. Could you provide interaction times for the RL trained agent with "check-again" strategy? If it fails to improve RL (well-)trained agents, how could we further increase the performance by increasing interaction horizons? Also, please remove the centering in Fig 5 (left), and remain the original interaction times.
3. Since the observation history is only recent 3 steps, how does the agent handle long-term dependencies when the interaction horizon is increased? And, under this setting, how does "check-again" prompting different to Reflexion [1]?
4. In L127, the paper mentions that prior works "usually scale the number of thinking tokens at each step" without reference. What are the prior works? Could you compare current TTI with these prior works under prompting and RL training?
5. **Open Question:** How should TTI be applied to other tasks when some tasks require high costs for interaction and with non-recoverable states (e.g., robotics, games, financial decision making)? If there is a place for thinking longer to be effective, how should TTI co-exist with thinking longer?

*If the authors provide evidence addressing Q1–Q4, corresponding rating score could increase.*

[1] Reflexion: Language Agents with Verbal Reinforcement Learning. NIPS 2023.

**Ethical Concerns:**

["NO or VERY MINOR ethics concerns only"]

**Final Justification:**

I find it highly difficult to carry out additional experiments to further address my concerns. However, I do not believe this work deserves rejection, so I will leave my current score of 4 as it is.

**Limitations:**

The authors acknowledge domain restriction to web-navigation and compute overhead of long horizons. However, the paper does not discuss the limitations of TTI in terms of generalization beyond web browsing. The application of overly long interaction horizons to crucial tasks with high costs for interaction and non-recoverable states (e.g., robotics, games, financial decision making) could be risky. Also, interacting with users in real-world applications could be a feasible scalable dimension. Overall, these limitations could block the applicability of TTI to many real-world tasks.

**Quality:**

3

**Strengths And Weaknesses:**

### Strengths

1. **Clear Writing**: The paper is well-structured, with a logical flow from motivation to method design to empirical results. Figures and tables are clear and informative.

2. **Well-Designed Preliminary Experiments**: The curriculum design is methodologically sound, with careful ablations on the effects of prompting towards TTI and acting longer. Many experiment details are provided, which is helpful for reproducibility.

3. **Detailed Case Studies**: The paper includes insightful case studies that illustrate how the agent learns to balance exploration and exploitation, as well as the types of errors it makes.

### Weaknesses

1. **Novelty of TTI**: Though the experiments are thorough with authentic RL curriculum, the core idea of scaling interaction horizons is intuitive and not novel. Besides a boarder demonstration of TTI in UI-TARS-1.5 [1], [2,3] have already shown that longer interaction horizons can improve agent performance substantially.

2. **Limited Interaction Horizon**: The maximum interaction horizon of 20/30 steps is still relatively short for complex tasks. While the paper shows that TTI improves performance on current benchmarks, it does not explore how much horizons could be scaled further like [1].

3. **Lack of Metrics/Baselines in RL Training**: The paper does not report how the thinking tokens change during RL training, along with the change of interaction times. Also, RL training to incentivize longer thinking (e.g., mixing reasoning data in agent RL) is not compared with RL training for TTI. This makes it hard to assess the effectiveness of the RL training in improving interaction horizons compared to improving thinking lengths. Most other open-source baselines are from other model backbones, limiting insights into the effectiveness of TTI.

4. **Too Compact Layout**: The paper is quite compact, with many figures and tables crammed into a small space, and still large amount of details are buried in the appendix. It is recommended to summarize the case studies instead of showing all the depiction. Training details such as data generation pipeline should be described more clearly as a part of the method.


[1] [UI-TARS-1.5 Blog](https://seed-tars.com/1.5/).
[2] CodeT: Code Generation with Generated Tests. ICLR 2023.
[3] Reflexion: Language Agents with Verbal Reinforcement Learning. NIPS 2023.

---

> ### Author Rebuttal · Authors · 2025-07-30
>
> Thank you for highlighting our work's clarity, careful experiments, and empirical insights. To address your questions regarding evaluation, **we add new results showing how per-step thinking changes during TTI training** and **new baselines that use RL to scale per-step thinking**. These results further showcase how TTI extends interaction and outperforms approaches that only scale per-step reasoning. Below, we elaborate on the findings and also discuss questions around related work, novelty, and broader applicability.
>
> ---
> ## Question 1
> > in Fig 4, after 400 steps, the agent (h=20) stop increasing overall interaction horizon but still improves the performance. Could you clarify this and the saying in L248?
>
> Fig 4 shows that interaction length saturates after 400 steps for h=20 agent, yet performance continues to rise. This is not a contradiction, rather, the agent learns to use its interaction budget more effectively, choosing better actions and exploring more strategically (e.g., searching for the correct item instead of clicking randomly). This motivates why we want to train agents with longer horizons **during later stages of training**.
>
> As for L248, it is meant to motivate why we want to use shorter horizons **during early stages of training**. By "worse performance stemming from noisy credit assignment and slower learning", we are referring to the initial phase of the h=20 agent (red dotted line in Fig 4 left), where the performance first decreases because h=20 agent has not yet grasped the basic abilities to solve tasks efficiently and associates random exploration moves with high rewards. Employing an adaptive horizon schedule via TTI can prevent the initial decrease and improve overall learning efficiency. We'll add this clarification to Sec 5.1.
>
> > provide the thinking lengths
>
> As suggested, we add two sets of results to show how thinking token changes during RL training. We show tables here but will include figures in the paper.
>
> **For WebArena (Fig 4),** we compute the average token in the "Thought" section for h=5/10/20.
> |Per-Step Thinking Token|Train Iter 2|4|6|8|10|
> |-|-|-|-|-|-|
> |h=5|124.2|143.1|142.5|171.7|181.2|
> |h=10|130.6|129.4|128.9|118.1|117.8|
> |h=20|116.8|105.7|115.2|104.6|94.3|
>
> For the h=5 agent, per-step reasoning increases over training, likely because the agent must solve the task within a limited number of steps, requiring more deliberate decisions. In contrast, h=20 agent relies more on extended interaction and shows a decrease in per-step reasoning. This aligns with our findings on large-scale WebVoyager (see below).
>
> **For WebVoyager results (Fig 5),** we show the following:
> |Per-Step Thinking Token|Train Iter 2|4|6|8|10|12|
> |-|-|-|-|-|-|-|
> |TTI|129.57|112.28|117.55|107.64|110.82|101.34|
>
> As training goes, the agent's trajectory grows longer (see Fig 5 left), but the number of tokens appearing in per-step reasoning actually becomes smaller. This implies that our agent is automatically learning to **trade-off interaction for per-step compute in order to attain higher performance**, and perhaps prevents any issues with overthinking each step.
>
> ---
> ## Question 2
>
> > In Fig 6, the "check-again" strategy seems to be less effective to increase the trajecory lengths. Could you provide interaction times for the RL trained agent with "check-again"? If it fails to improve RL (well-)trained agents, how could we further increase the performance?
>
> As suggested, we provide the interaction times for RL agent with check-again.
> |# Interaction Steps|No Re-check|1 Check|2 Checks|3 Checks|
> |-|-|-|-|-|
> |Before Training|6.03|8.24|10.67|12.25|
> |Iter 5|9.01|10.49|11.95|13.61|
> |Iter 10|10.53|11.33|11.84|12.5|
>
> As you pointed out, the increase in interaction steps is smaller for well-trained agents than baseline agents. Fig 6 also shows that the benefit of check-again diminishes as the agent learns to self-reflect during TTI training. This shows TTI agents learn to internalize re-checking and interaction scaling behavior via curriculum RL.
>
> To further increase interaction horizons for well-trained agents, we could consider new directions like hierarchical exploration (e.g., setting and pursuing sub-goals within a single rollout). These are future work beyond the current scope of TTI. We'll add a discussion to encourage exploring this in the community.
>
> ---
> ## Weakness 3
>
> > RL training to incentivize longer thinking is not compared with RL training for TTI
>
> As suggested, we add new baselines that use RL to scale per-step reasoning for the WebVoyager experiments. We tried two methods: (a) using a more detailed instruction for the "Thought" section, or (b) asking the agent to double check its action at each step by outputting two "Thought…Action…" blocks. We use a fixed horizon of 10.
> |Method|Success Rate(%)|Per-Step Thinking Token Before Training|After Training|
> |-|-|-|-|
> |Baseline (Prompt Only)|55.8|-|-|
> |Fixed h=10|59.1|131|110|
> |Fixed h=30|45.2|131|119|
> |RL for Scaling Per-Step Thinking (a)|60.5|131|197|
> |RL for Scaling Per-Step Thinking (b)|50.0|131|216|
> |TTI (Ours)|**66.5**|131|101|
>
> Both (a) and (b) increase per-step thinking. (a) improves over the fixed h=10 baseline but still underperforms compared to TTI. This suggests that simply thinking more per step is not enough; **TTI's ability to scale interaction allows the agent to actively explore the environment and gather information critical to task success**, leading to stronger performance. Also, method (b) seems to hurt performance likely because it adds redundancy rather than improving exploration or decision quality. We’ll include the new results in the paper.
>
> ---
> ## Question 3
>
> > observation history is only recent 3 steps, how does the agent handle long-term dependencies
>
> As noted in Sec 3, the number of **web page observations** is kept to 3; our agent still has full access to **all past actions**. This action history can serve as a compressed task "memory". Also, TTI implicitly teaches the agent to revisit pages, backtrack, and gather missing information dynamically to improve long-term dependencies (see case studies in Sec 6.2).
>
> ---
> ## Question 4
>
> > L127 mentions prior works "usually scale the number of thinking tokens at each step" without reference. What are the prior works? Could you compare TTI with these prior works under prompting and RL training?
>
> To the best of our knowledge, prior to submission, there were no **multi-step agent** works that systematically scale per-step thinking tokens during training (most focus on single-turn reasoning tasks like math). Recent work like Kimi K2 explores this, but it appeared much after our submission. That said, we already compare to baselines that scale thinking tokens:
> 1. Muennighoff et al. for test-time budget forcing (see Sec 4.2)
> 2. Claude, OpenAI models, and others trained for internal CoT scaling (see Tables 3, 4)
> 3. New RL baselines for scaling per-step reasoning, which we include in this rebuttal (see response to W3).
>
> If we are missing prior work, please let us know and we are happy to compare!
>
> ---
> ## Weakness 1
>
> > core idea of scaling interaction horizons is intuitive and not novel
>
> While we agree that TTI's core idea is intuitive, we view its simplicity as a strength, not a weakness. Crucially, our idea is still novel, as prior work (e.g., Reflexion, CodeT) has used longer interactions primarily at **test time**, either for evaluation or to inject human priors via prompting. In contrast, our goal is to build a systematic curriculum over interaction steps that **trains** agents via RL to acquire the ability to leverage longer interaction horizons to improve performance. For prior approaches, such ability is not explicitly optimized during training. While scripted retries or prompting tricks can still tackle a few extra tasks with longer test-time horizons, their agents **lack the capacity to adaptively scale interaction**, which TTI agents are capable of.
>
> ---
> ## Weakness 2
>
> > 20/30 steps is still relatively short for complex tasks
>
> We note that a 30-step horizon is already 2-5x longer than typical task lengths in WebArena (~6 steps), Mind2Web (~8 steps), and UI-TARS-1.5 training data (~15 steps), and aligns with the standard 30-step limit used in most benchmark frameworks. This provides sufficient room for exploration while remaining compute-efficient. We agree that longer horizons may be necessary in other domains and we encourage researchers to adapt step limits to their specific settings.
>
> ---
> ## Open Questions
>
> > How should TTI be applied to other tasks that require high costs for interaction and with non-recoverable states? If there is a place for thinking longer to be effective, how should TTI co-exist with thinking longer?
>
> We think the following strategies might be worth exploring:
> 1. Simulated interaction or planning can be used to reduce the number of costly real-world steps. For example, we can develop hierarchical TTI that differentiates between cheap simulated interactions and expensive physical actions, allocating horizon accordingly
> 2. We can also develop risk-aware curricula that gradually expose agents to higher-cost actions, e.g., in robotics and in several games, you have the option of "going back and retrying", meaning there is a possibility of developing risk aware curricula
>
> Akin to how long CoT reasoning is not broadly applicable to every domain, long interaction lengths are also not, but this does not undermine the need for long CoT reasoning models.
>
> Furthermore, our results suggest that longer per-step thinking is best suited for local optimization, validation, or final decision verification, whereas external interaction is more effective for exploration, gathering missing information, and adapting to partial observability. We believe an important future direction is *hybrid test-time scaling*, where agents learn to allocate their compute budget between internal reasoning and external action based on the environment’s cost and the agent's uncertainty.

---

> > ### Comment · Reviewer_nVXA · 2025-08-04
> >
> > Thanks for your rebuttal. I will keep my score as it is.

---

> > > ### Author Response · Authors · 2025-08-04
> > >
> > > Thank you for your thoughtful review and for engaging with our rebuttal. We’re glad the additional results addressed your initial questions. If there’s anything else we can clarify or provide that might help you consider raising your score, we’d be happy to do so. We appreciate your time and consideration!

---

> > > > ### Comment · Reviewer_nVXA · 2025-08-05
> > > >
> > > > I find it highly difficult to carry out additional experiments to further address my concerns. However, I do not believe this work deserves rejection, so I will leave my current score of 4 as it is.

---

### Decision · Program_Chairs · 2025-09-17

**Decision:**

Accept (poster)

**Comment:**

This submission introduces a curriculum-based online reinforcement learning method (test time interaction, tti). Its goal is to scale the interaction of agents rather than only increasing per-step reasoning. The authors argue that longer interaction horizons allow agents to gather more information, adapt strategies, and achieve improved performance on complex tasks. Empirical evaluations on WebVoyager and WebArena show that a Gemma-3 12B model fine-tuned with tti achieves new sota results among open-source agents. Ablation studies and case analyses also show that tti enables adaptive exploration, backtracking, and dynamic re-planning. This helps provide a complementary scaling axis to traditional "thinking longer" approaches.

Strengths of this submission:
1. reviewers kwcd, 13jF, and nVXA believe that tti demonstrates consistent gains over baselines on two challenging web benchmark. The experiments also cover ablations, case studies, and new baselines comparing per-step reasoning versus interaction scaling.
2. reviewers kwcd, 13jF like the technical idea presented. It identifies interaction scaling as an underexplored dimension for test-time scaling, showing its practical importance to existing approaches.
3. reviewers nVXA and kwcd consider that this submission is well presented.

Weaknesses of this submission:
1. Safety and applicability concerns: reviewer GJf2 strongly believes that scaling interaction steps can introduce risks in real environments, such as harmful, risky, or irreversible behaviors, which are insufficiently discussed.
2. reviewer nVXA finds that the intuition that longer interaction horizons improve agent performance has been observed in prior works, and the contribution may be incremental.
3. reviewers GJf2 and nVXA also mention that the curriculum design for scaling horizons is largely heuristic, lacking theoretical grounding or cross-domain validation.

Despite concerns about safety, the paper provides a compelling contribution by establishing interaction scaling as a complementary and underexplored axis for improving agent performance. The experiments are relatively strong, with ablations, comparisons against strong baselines, and sota results on two benchmarks. While safety concerns raised by one reviewer GJf2 are valid for deployment in high-risk domains, they do not undermine the technical contributions and empirical insights of this work in the evaluated setting.

During the rebuttal, the authors added new analyses addressing several reviewers' concerns. They provided per-step thinking token trends, new RL baselines for scaling per-step reasoning, compute cost breakdowns, and per-domain training curves, which addressed questions from reviewers nVXA, kwcd, and 13jF. However, reviewer GJf2 maintained concerns about potential safety risks and the broader applicability of tti, arguing that longer interaction horizons could lead to unsafe behaviors in risky domains; the authors responded by clarifying that tti’s RL objective discourages harmful actions and discussing safe RL extensions and sandboxed environments, though these points did not fully convince reviewer GJf2. The other three reviewers expressed satisfaction with the clarifications and maintained their positive evaluations, supporting acceptance.

All factors considered, this submission is given an acceptance recommendation. The major rational of this decision lies in 1) concerns about long-horizon reasoning producing risky behaviors do not necessarily imply a negative impact on the contribution of the proposed technique, and 2) the perspective provided in this submission is attractive though not perfect, and the ac does not find strong reasons to reject this submission.